# Time-resolved structure-function coupling in brain networks

Zhen-Qi Liu [1,3], Bertha Vázquez-Rodríguez [1,3], R. Nathan Spreng [1], Boris C. Bernhardt [1], Richard F. Betzel [2] & Bratislav Misic [1✉]

The relationship between structural and functional connectivity in the brain is a key question in systems neuroscience. Modern accounts assume a single global structure-function relationship that persists over time. Here we study structure-function coupling from a dynamic perspective, and show that it is regionally heterogeneous. We use a temporal unwrapping procedure to identify moment-to-moment co-fluctuations in neural activity, and reconstruct time-resolved structure-function coupling patterns. We find that patterns of dynamic structure-function coupling are region-specific. We observe stable coupling in unimodal and transmodal cortex, and dynamic coupling in intermediate regions, particularly in insular cortex (salience network) and frontal eye fields (dorsal attention network). Finally, we show that the variability of a region's structure-function coupling is related to the distribution of its connection lengths. Collectively, our findings provide a way to study structure-function relationships from a dynamic perspective.

[1] McConnell Brain Imaging Centre, Montréal Neurological Institute, McGill University, Montréal, Canada. [2] Psychological and Brain Sciences, Program in Neuroscience, Cognitive Science Program, Network Science Institute, Indiana University, Bloomington, IN, USA. [3] These authors contributed equally: Zhen-Qi Liu, Bertha Vázquez-Rodríguez. ✉email: bratislav.misic@mcgill.ca

The brain is a network of anatomically connected neuronal populations. Inter-regional signaling via electrical impulses manifests as patterns of organized co-activations, termed "functional connectivity". The coupling between structural connectivity (SC) and functional connectivity (FC) is a fundamental feature that reflects the integrity of neural signaling[1]. Historically, most studies have focused on static structure-function coupling over the course of a whole scanning session[2].

However, over the past decade functional connectivity is increasingly conceptualized as a dynamic process[3–5]. Functional connectivity patterns display time-resolved fluctuations that are non-random[6–10], highly organized[11–14], individual-specific[15], related to behavior[16,17], and evolve over the lifespan[18]. As a result, structure-function coupling should fluctuate over multiple time-scales. Indeed, multiple studies have reported evidence of dynamic structure-function relationships over the course of single recording sessions[19,20], and over more protracted periods, including early childhood and young adult neurodevelopment[21,22].

Importantly, previous studies on dynamic structure-function coupling worked under the assumption that structure-function relationships are uniform across the brain. Recent research suggests that structure-function coupling is regionally heterogeneous, such that structural and functional connectivity profiles are closely related in sensory (unimodal) cortex, but gradually decouple in transmodal cortex[21,23–25]. The systematic decoupling of structure and function along this unimodal-transmodal gradient is thought to reflect differentiation in micro-architectural properties[2,26,27], including molecular, cellular, and laminar differentiation[21,23,28,29]. Indeed, computational models that implement regionally heterogeneous dynamics using micro-architectural properties make more accurate predictions of functional connectivity from structural connectivity[30–33].

How do regional patterns of structure-function coupling fluctuate moment-to-moment? We considered two alternative possibilities. One possibility is that structure-function coupling is greater in transmodal cortex. Several recent studies have shown that static structure-function coupling is lower in transmodal cortex compared to unimodal cortex[2,21,23,34]. Given that transmodal cortex engages in multiple polysensory functions and functional relationships, a plausible explanation could be that greater variability in time-dependent structure-function coupling ultimately averages out and appears as lower static structure-function coupling. Another possibility is that structure-function coupling is greatest in regions that are intermediate in the putative unimodal-transmodal hierarchy. Numerous evidence points to diverse cytoarchitecture and connectional fingerprints in insular cortex[35,36]. By participating in a diverse set of connections with multiple brain regions, the insula is thought to dynamically engage in multiple cognitive systems[37–39].

Here we derive time- and region-resolved patterns of structure-function coupling. We first estimate dynamic inter-regional co-fluctuation using a recently-developed temporal unwrapping method that does not require windowing[9,14]. We then reconstruct dynamic patterns of regional structure-function coupling and contextualize these patterns with respect to macroscale brain organization, including intrinsic networks, as well as functional and cellular hierarchies.

## Results
The results are organized as follows. We first reconstruct frame-by-frame co-fluctuation matrices from regional BOLD time-series[9,14]. We then apply a multilinear model to estimate regional time-series of structure-function coupling[23], before comparing regional fluctuations in structure-function coupling with large-scale intrinsic networks[40], cortical hierarchies[41], and cytoarchitectonic classes[42]. We also benchmark the extent to which dynamic fluctuations in structure-function coupling can be explained by topological and geometric embedding. Finally, we assess the correspondence between conventional (static) structure-function coupling and dynamic structure-function coupling. Data were derived from $N = 327$ healthy, unrelated participants from the Human Connectome Project (HCP)[43]. Structural connectomes were reconstructed from diffusion MRI (dMRI). Static and dynamic functional connectivity were estimated from resting-state functional MRI (fMRI) (see *Materials and Methods* for detailed procedures). Analyses were performed using a network parcellation of 400 cortical nodes[44].

**Time-resolved structure-function coupling.** The temporal unwrapping procedure generates a node-by-node co-fluctuation matrix for each time point (Fig. 1a). We then use a multilinear regression model to predict the co-fluctuation profile of every node from its structural connectivity profile[23,45]. The model was fitted separately for each time point (Fig. 1c). The regression model incorporates multiple computational models of cortical communication[1,46]: (1) Euclidean distance, (2) shortest path length, and (3) communicability (Fig. 1b). Euclidean distance embodies the notion that proximal neurons may exchange information more easily, and is consistent with navigation-like communication[47]. Shortest path length is a statistic that embodies centralized routing-like communication[48], while communicability is a statistic that embodies decentralized diffusion-like communication[49,50]. Note, however, that there exist multiple alternative statistics that measure the capacity of the network to transmit information, and the three chosen measures constitute a subset of that wider space[51]. All models were fitted independently for each individual participant.

The multilinear model allows us to quantify regional structure-function coupling across time. For each brain region $i$ and time point $t$, we measure the goodness of fit using the coefficient of determination $R_{i,t}^2$ between the predicted and the empirical functional profile (Fig. 1c). A value near 1 indicates strong coupling between the structural and functional profiles for the $i^{th}$ node at time $t$. These coefficients of determination are then assembled into a node × time structure-function coupling matrix. The procedure was carried out separately for each individual in the sample.

Figure 2a shows the Pearson correlations between dynamic structure-function coupling maps and the static structure-function coupling map reconstructed using the whole time-series. The coefficients span a wide distribution, encompassing both positive and negative values. A distribution of coefficients is mathematically expected given that the method is measuring the relationship between dynamic functional connectivity and static structural connectivity. In the present report, we further analyze how dynamic functional connectivity around a static structural connectivity reference yields fluctuations in structure-function correlations, and we map these fluctuations to the cortical hierarchy. Figure 2b shows the relationship between two alternative methods for estimating regional structure-function coupling. The abscissa shows structure-function coupling values estimated using the multilinear model described above, while the ordinate shows the same values estimated using the method described by Baum and colleagues[21]. The latter, which we term "Spearman rank coupling", estimates structure-function coupling as the Spearman rank correlation between the structural and functional profiles of each node. The principal strength of the method is that it does not make arbitrary assumptions about which predictors to include; the principal weakness is that the correlation can only be computed between pairs of regions that

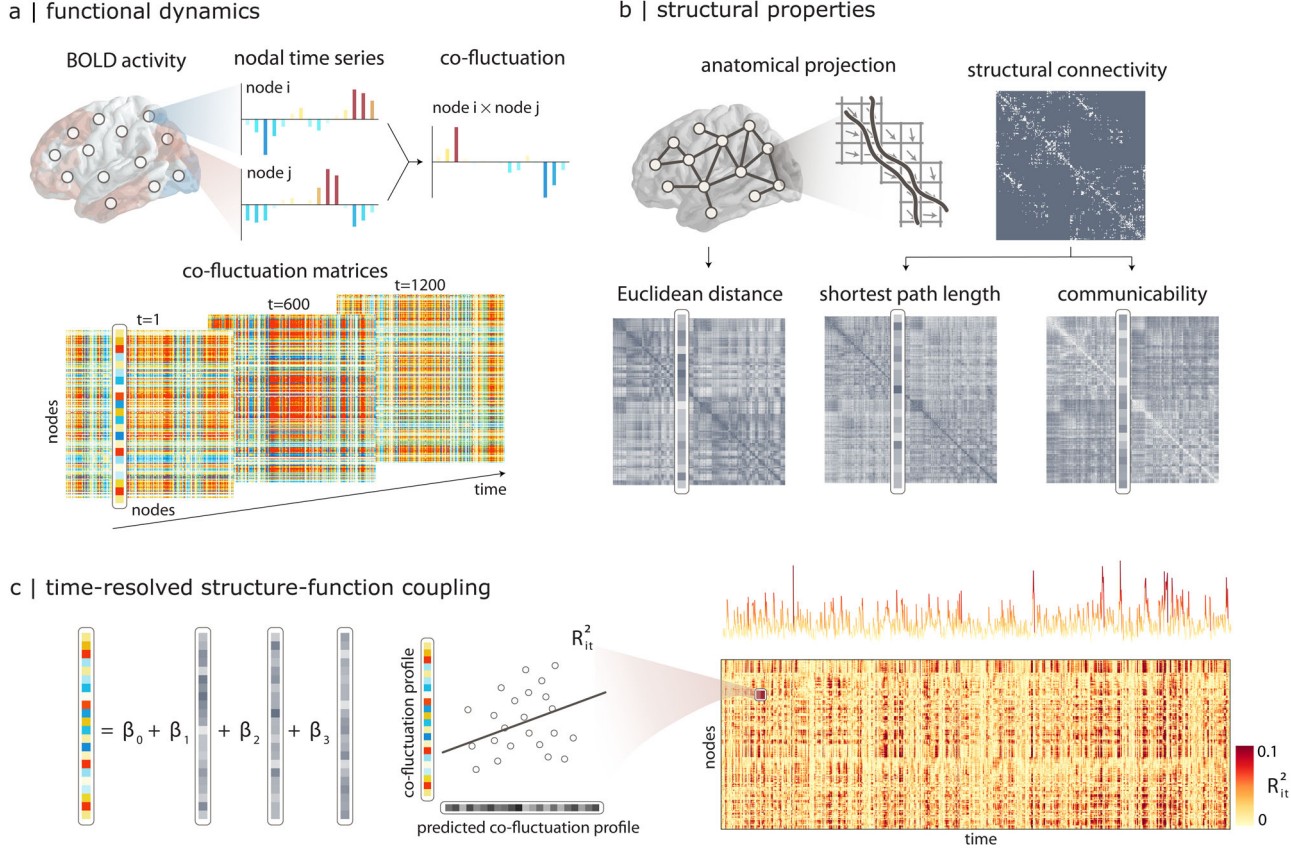

**Fig. 1 Time-resolved structure-function coupling. a** The co-fluctuation of two brain regions $i$ and $j$ is calculated as the element-wise multiplication of the two z-scored fMRI BOLD activity time-series. The points of this time-series can be represented as one element in a co-fluctuation matrix. **b** Pairwise structural relationships are derived from structural connectivity networks reconstructed from diffusion MRI, including Euclidean distance between node centroids, shortest path length and communicability. **c** A multilinear regression model is used to predict a region's co-fluctuation profile from its structural profile, using Euclidean distance, path length and communicability as predictors. The resulting coefficient of determination ($R^2_{i,t}$) indicates how well the structural connectivity profile predicts the functional connectivity profile for a particular brain region $i$ at a particular time point $t$. The procedure generates a region × time matrix that captures the fluctuation of structure-function coupling for individual regions across time. The time-series shows time-resolved fluctuations in mean $R^2$.

have an underlying structural connection, potentially missing out on biologically-important dyadic relationships. Importantly, the two methods are positively correlated ($r = 0.22$), suggesting that, while the two methods offer qualitatively similar perspectives on structure-function coupling, they are not perfectly correlated, potentially because one is sensitive to direct monosynaptic relationships while the other also takes into account polysynaptic relationships.

Figure 2c shows the mean structure-function coupling $R^2$, while Fig. S1 shows the contribution of individual predictors. The relationship between $R^2$ and co-fluctuation amplitude is shown in Fig. 3. To quantify the variability of structure-function coupling across time, we compute—separately for each participant—the coefficient of variation of $R^2$ across time ($cv(R^2)$). The coefficient of variation is the ratio of the standard deviation of $R^2$ to the mean of $R^2$. It is a standardized measure of dispersion of $R^2$ values about the mean that captures the variability in structure-function coupling across time. In other words, $cv(R^2)$ allows us to compare the variability of structure-function coupling time-series that have different means. Figure 2d shows that $cv(R^2)$ is regionally heterogeneous and appears to be greatest in insular cortex, frontal eye fields, medial prefrontal, and medial occipital cortex. In the following section, we analyze this pattern in greater detail.

**Hierarchical organization of dynamic structure-function coupling.** We next consider how patterns of dynamic structure-function coupling reflect different features of cortical organization. Specifically, we focus on three widely studied cortical annotations, including the unimodal-transmodal principal functional gradient[41], intrinsic functional networks[44], and cytoarchitectonic classes[42]. In each case, we compute the mean coefficient of variation of structure-function coupling. Figure 3b shows exemplar time-series of structure-function coupling for nodes in insular and parietal cortex, exhibiting distinct variability patterns. Figure 3c–e shows that brain regions that occupy intermediate positions in the cortical hierarchy tend to display the most dynamic fluctuations in structure-function coupling. Specifically, we find the most variable fluctuations in the middle of the unimodal-transmodal hierarchy (classes 4-6), corresponding to the ventral attention/salience network and the insular cortex in the Yeo and Von Economo atlases, respectively, as well as the frontal eye fields, corresponding to the dorsal attention network. These observations are confirmed using spatial autocorrelation-preserving null models to test the null hypothesis that $cv(R^2)$ is uniform across the brain. The tests reveal significantly greater $cv(R^2)$ in intermediate positions of the unimodal-transmodal hierarchy (Fig. 2). Namely, the insular cortex and frontal eye fields, intermediate in the unimodal-transmodal hierarchy, have

**Fig. 2 Dynamic structure-function coupling. a** Correlations between regional patterns of static and dynamic structure-function coupling. **b** Correlations between dynamic structure-function coupling estimated using a multilinear model[23] versus coupling estimated using an alternative Spearman rank correlation method[21]. Scatter color and size represent the density. **c** Mean time-resolved structure-function coupling over time (left) and its mean over subjects (right). **d** Coefficient of variation of structure-function coupling across time (left), and its mean over subjects (right).

the most variable structure-function coupling, while unimodal and transmodal cortex have more stable structure-function coupling.

**Relating static and dynamic structure-function coupling.** In the previous section, we considered how structure-function coupling fluctuates around the mean. We next ask: how closely do dynamic patterns of structure-function coupling reflect static structure-function coupling? To address this question, we systematically compare the dynamic and static case. Taking into account all time points in the dynamic case, we compute (a) the proportion of time points for which dynamic coupling is greater than static coupling ("dynamic > static"), (b) how similar the dynamic patterns are to the static pattern ("bias") and, (c) how tightly scattered the dynamic patterns are relative to the static pattern ("variance") (Fig. 4a).

We find that regions intermediate in the unimodal-transmodal hierarchy, corresponding to the insular cortex, tend to have relatively greater dynamic than static coupling compared with other groups in the hierarchy (up to 0.5, Fig. 4b). These regions also have the closest correspondence between dynamic and static coupling (Fig. 4c) and the lowest dynamical variance around the static case (Fig. 4d). Altogether, these results suggest that the relationship between dynamic and static coupling is not uniform across the brain, but strongly depends on the region's position in the putative unimodal-transmodal hierarchy, with the closest correspondence between static and dynamic coupling observed in

the middle of the hierarchy. Taken together with the results from the previous section, we reveal an interesting property about areas that are intermediate in the hierarchy, such as insular cortex and frontal eye fields. Namely, intermediate areas display the greatest overall fluctuations relative to the mean, but over time tend to follow and converge with static coupling.

**Spatial and topological determinants of dynamic structure-function coupling.** We finally seek to understand how dynamic local structure-function coupling depends on geometric, anatomical and functional embedding. Given that the unimodal-transmodal hierarchy possibly reflects a continuous gradient of connection lengths[52–54], we ask whether dynamic structure-function coupling also reflects the distribution of connection lengths that a region participates in. Figure 5a shows the map of mean connectivity distance for each region[53,54]. We find that areas with very short and very long connection lengths tend to have more stable coupling, and areas with intermediate connection lengths tend to have more variable coupling. Figure 5b shows correlations between dynamic structure-function coupling and multiple measures of structural and functional network embedding, including betweenness, clustering and degree. Robust correlation analysis (biweight midcorrelation and percentage bend correlation[55]) suggests significant and stable correlations with structural degree (−0.1667; −0.1722; −0.1605), mean edge length (−0.1977; −0.1987; −0.1936), and functional strength (0.235; 0.1956; 0.1803). Altogether, these results suggest that the dynamic

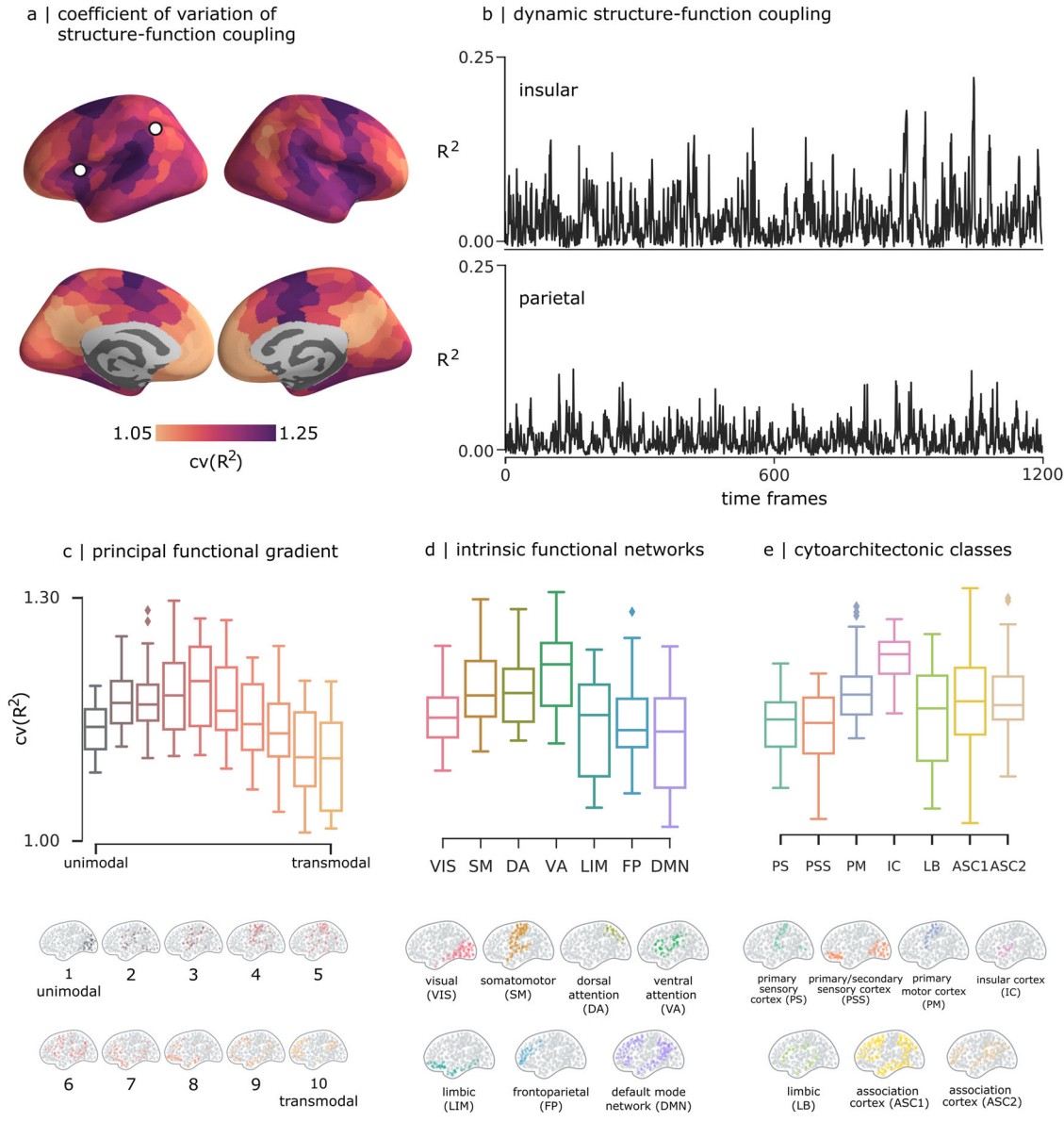

**Fig. 3 Relationship with cortical hierarchies. a** Coefficient of variation of structure-function coupling, averaged over all participants. **b** Time-series of regional structure-function coupling shown for one region in parietal cortex (left) and one region in insular cortex (right) from one randomly selected participant. The mean coefficient of variation is displayed for three types of cortical annotations: **c** 10 equally-sized bins of the principal functional gradient[41], **d** intrinsic functional networks[40], and **e** von Economo cytoarchitectonic classes[42].

nature of structure-function coupling in "middle hierarchy" regions potentially originates from their connection length distribution.

Interestingly, when we compute the group-average similarity of inter-regional structure-function time-courses (i.e., how similar are inter-regional fluctuations in structure-function coupling), we find a comparable relationship with Euclidean distance (Fig. 5d). Namely, regions that are physically close together and far apart tend to display similar fluctuations in structure-function coupling, and regions that are at intermediate distances from one another tend to display dissimilar fluctuations in coupling. Finally, we compare the similarity of structure-function coupling between regions with the structural and functional connectivity between those regions. We find that the mean similarity of structure-function coupling is greater for areas that are structurally connected than areas that are not ($t(79798) = 80.95$, $p < 0.001$) (Fig. 5e). Likewise, mean similarity of structure-function coupling is greater for areas that participate in the same

intrinsic networks than those that are in different networks ($t(79798) = 45.34$, $p < 0.001$) (Fig. 5e). In other words, coordinated patterns of dynamic structure-function coupling are—as expected—driven by inter-regional structural and functional connectivity.

## Discussion
Emerging theories emphasize dynamic functional interactions that unfold over structural brain networks[3]. Here, we study time- and region-resolved patterns of structure-function coupling. We find that dynamic coupling patterns reflect cortical hierarchies, with the most dynamic fluctuations in the insula and frontal eye fields. These graded patterns of dynamic coupling reflect the topological and geometric embedding of brain regions.

Our results build on recent work showing that structure-function coupling is not uniform across the brain, but highly region-specific[21,23,24,32]. These studies have consistently

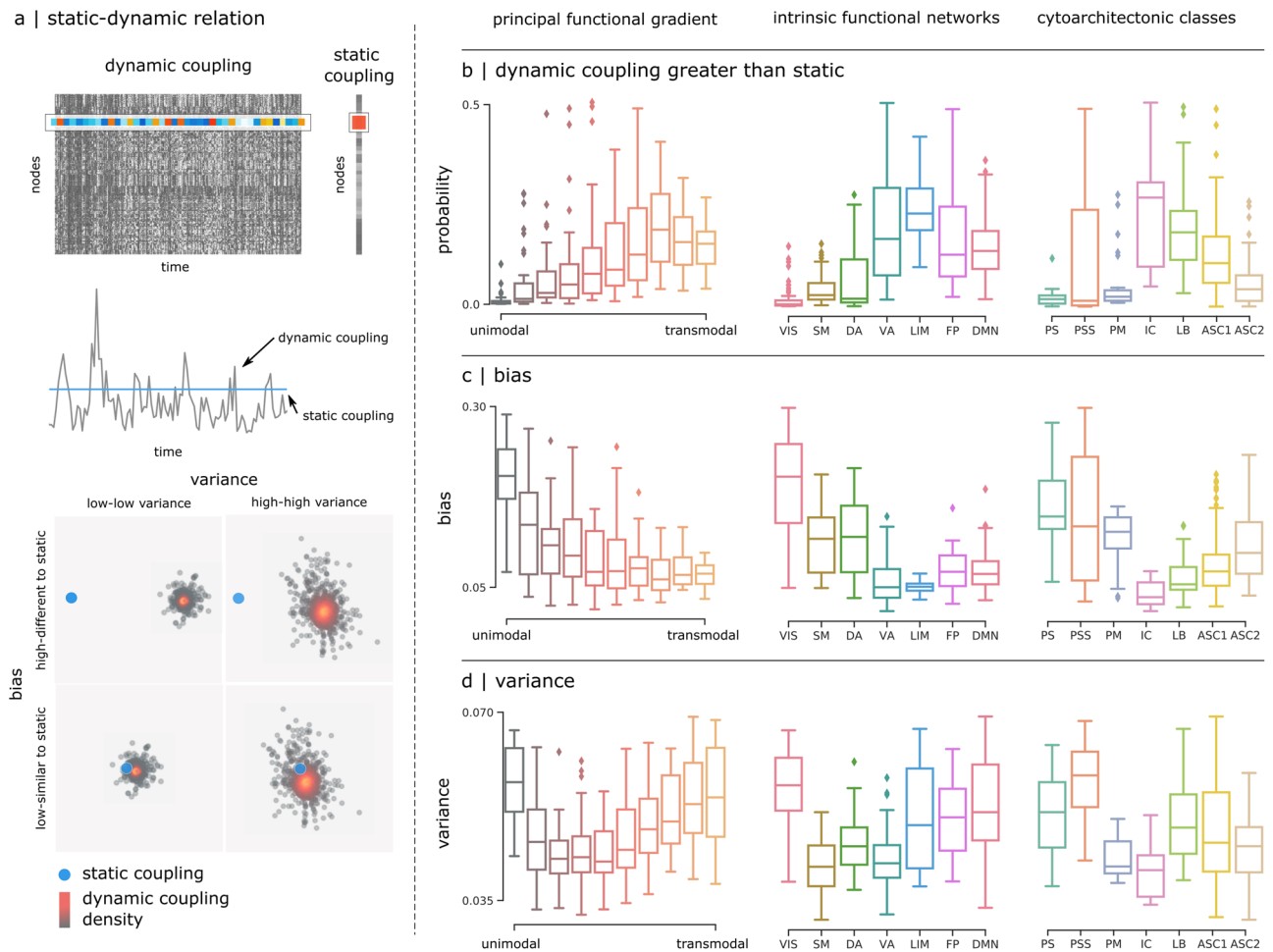

**Fig. 4 Relating static and dynamic structure-function coupling. a** Top: static structure-function coupling is estimated using the functional connectivity matrix derived from the whole resting-state time-series[23], and compared with dynamic coupling. The dynamic structure-function coupling of node $i$ corresponds to the $i$th row of the dynamic coupling matrix, while the static coupling corresponds to the $i$th element of static coupling vector. Middle: dynamic values represented as a time-series (black line) that fluctuates around the single static coupling value (blue line). Bottom: dynamic coupling values are represented as a scattered distribution of points (black) around the static coupling value (blue point). The two are compared in different cortical annotations using three summary statistics: **b** the probability of having a larger dynamic coupling value compared to the static coupling, **c** the bias, and **d** the variance of the dynamic coupling to reproduce the static values.

demonstrated that structure-function coupling is graded, with strong coupling in unimodal cortex and weak coupling in transmodal cortex. By applying a temporal unwrapping method to estimate functional co-fluctuation patterns from moment-to-moment, we show that structure-function coupling is not only regionally heterogeneous, but also highly dynamic[19]. Namely, we find that the extremes of the putative unimodal-transmodal hierarchy display more stable structure-function coupling, while regions intermediate in the hierarchy display more sizable fluctuations.

Interestingly, the most dynamic fluctuations were observed in insular cortex and frontal eye fields. In concert with the anterior cingulate and dorsolateral prefrontal cortices, the insula forms the ventral attention or salience network, which supports the orienting of attention to behaviorally-relevant stimuli, including sensory and autonomic signals related to the internal milieu[35,36]. By participating in a diverse set of interdigitated connections with multiple brain regions, the insula is thought to dynamically coordinate communication among multiple cognitive systems[37–39]. In particular, the posterior portion of the insula displays prominent functional connectivity with sensory regions, while the anterior portion is primarily connected with frontal

areas involved in higher cognitive function[39,56]. In a similar vein, the frontal eye fields constitute a key node in the dorsal attention network, involved in biasing attention towards top-down goals and information foraging[57,58].

Aligning these two findings, we observe a common functional theme of regions on the interface between higher-order hetero-modal cognition and primary perceptual and internal states. We speculate that the greater variability in local structure-function coupling in the insula and frontal eye fields delineates a potential mechanism by which signals are flexibly routed through these unique cortical hubs across wide domains. These "middle hier-archy" regions must engage in particularly broad coordination patterns, integrating ongoing unimodal information processing with the more sustained and extended operations in heteromodal cortex. This information is likely weighted by salience and goal relevance, while also allowing novel ongoing sensory information to gain access to heteromodal cortex.

The graded nature of local structure-function coupling appears to be shaped by the geometric embedding of individual brain regions. Namely, we also find that regions with very short or very long connectivity distance tend to display stable coupling, while regions with intermediate connectivity distance, particularly

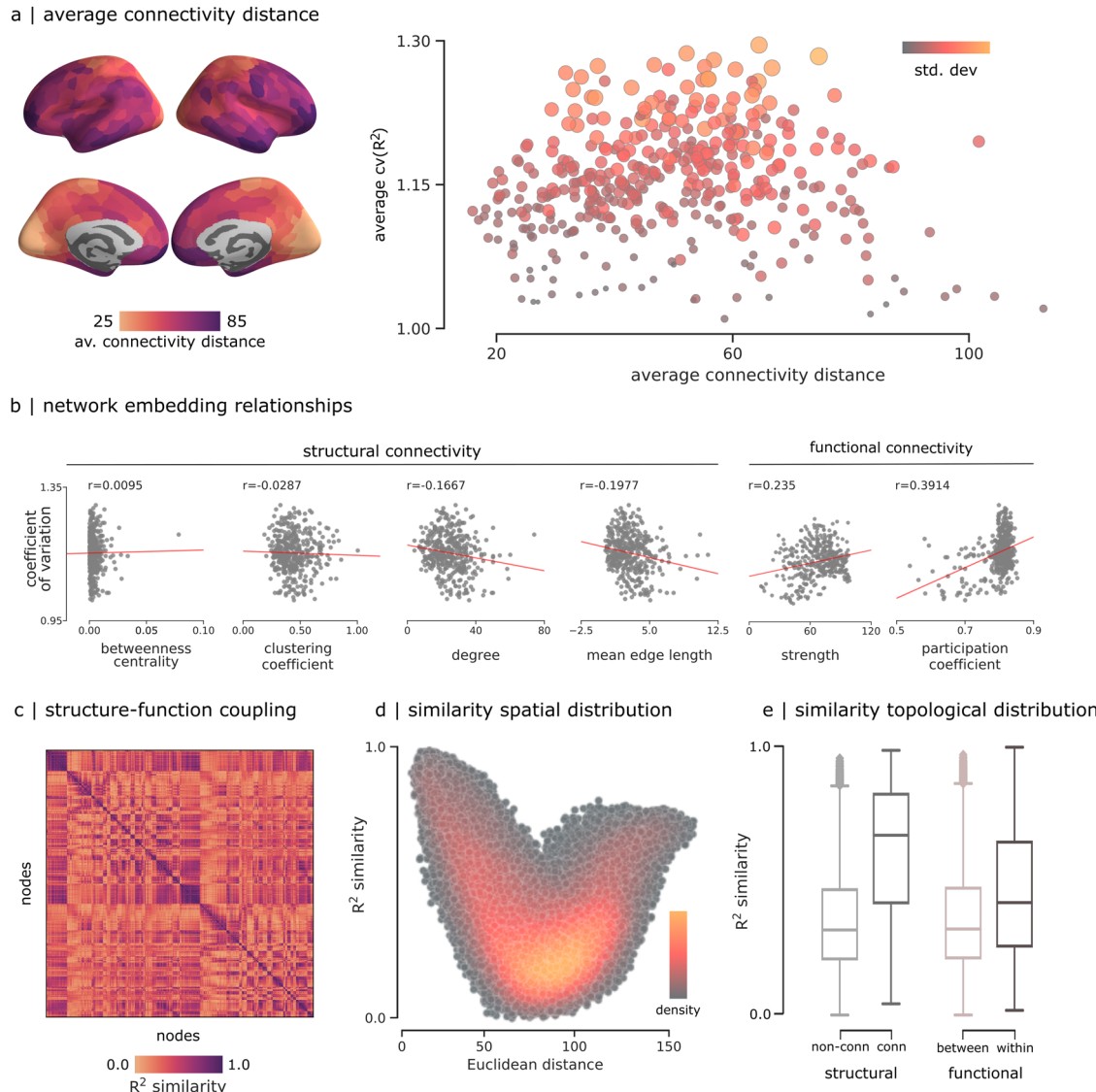

**Fig. 5 Spatial and topological determinants of structure-function coupling variability. a** Average connectivity distance calculated following[53], and correlated with the average coefficient of variation of the structure-function coupling from Fig. 3a. Scatter color and size represent the standard deviation. **b** Coefficient of variation of structure-function coupling compared to network embedding metrics derived from structural and functional networks. To account for the possible effects of outliers, we also estimated these relationships using the biweight midcorrelation ($r = 0.0031$; $-0.0349$; $-0.1722$; $-0.1987$; $0.1956$; $0.1647$) and percentage bend correlation ($r = -0.0152$; $-0.0359$; $-0.1605$; $-0.1936$; $0.1803$; $0.2980$)[55]. **c** $R^2$ similarity between pairs of nodes calculated as the Pearson correlation between pairs of regional structure-function coupling averaged across subjects (**d**) $R^2$ similarity correlated with Euclidean distance. Colormap shows the density of the scatter plot. **e** $R^2$ similarity values grouped by structural connectedness and functional intrinsic networks.

insular cortex and frontal eye fields, display more variable structure-function coupling. These findings resonate with a growing appreciation for how geometric relationships shape topological relationships in the brain[52,59–64]. In particular, physical separation from sensorimotor cortex is thought to correspond to graded variation in connectivity distance, culminating in predominantly long-range functional connectivity in association cortex[26,41,53,54,65]. The particular distribution of connection lengths that "middle hierarchy" regions participate in—leaning neither toward overly short- or long-range connectivity—may support flexible reconfiguration and participation in multiple systems[36–39,58], manifesting as variable structure-function coupling.

Our results build on a rapidly-developing literature on local structure-function relationships[2]. While traditional studies have focused on global structure-function relationships captured by a single forward model[45,66–69], numerous recent reports point to region-specific structure-function coupling patterns[21,23,24]. These structure-function relationships undergo extensive maturation and lifespan trajectories[21,70]. Interestingly, regional differences in structure-function coupling are correlated with micro-architectural variations, including intracortical myelin and cellular composition[21,23]. This suggests that local circuit properties —invisible to macroscale connectivity reconstructions—may additionally drive structure-function coupling[26]. Consistent with this notion, multiple modeling studies have recently shown that biophysical models constrained by regionally heterogeneous micro-architectural information, such as myelination, gene expression and neurotransmitter receptor profiles, make more accurate predictions about functional connectivity compared to

regionally homogeneous models[31–33]. How regional differences in micro-architecture shape moment-to-moment fluctuations in structure-function coupling remains an important question for future research.

The present results need to be interpreted with respect to multiple limitations. First, structural connectivity networks were reconstructed using diffusion weighted MRI, a method that is susceptible to systematic false positives and negatives[71–74]. Although the present findings are observed in individual participants and can be demonstrated using alternative methods, further development in computational tractometry is necessary. Second, the dynamics of the BOLD signal itself are influenced by multiple physiological confounds, including blood flow and respiration[75,76]. In the absence of concurrent measurements of cardiovascular and cerebrovascular factors, these results must be interpreted with caution. Likewise, it is important to note that there exist multiple alternative methods to quantify dynamic functional connectivity. We applied a recently-developed temporal unwrapping method that has been demonstrated to be robust to a wide range of methodological choices, including parcellation and global signal regression method, and are sensitive to individual differences[9,14]. The statistical properties of the underlying dynamic processes behind moment-to-moment functional dynamics of the human brain has been an area of active research for years[77–82], and the applicability of these methods to studying structure-function relationships is increasingly recognized[83,84].

Collectively, the present work identifies patterns of local structure-function coupling that are systematically organized across the cortex and highly dynamic. The temporal coupling of structure and function points towards a rich and under-explored feature of the brain that may potentially help to understand how functions and cognitive processes are flexibly implemented and deployed.

## Methods

**Data acquisition**. Structural and functional data were obtained from the Human Connectome Project (s900 release[43]). Scans from 327 healthy young participants (age range 22–35 years) with no familial relationships were used, including individual measures of diffusion MRI and four resting-state functional MRI time-series (two scans on day 1 and two scans on day 2, each of 15 min long). Data were processed following the procedure described in ref. [28,85].

**Structural network reconstruction**. Gray matter was parcellated into 400 cortical regions according to the Schaefer functional atlas[44]. Structural connectivity between regions was estimated for each participant using deterministic streamline tractography. First, the distribution of fiber orientation for each region was generated using the multi-shell multi-tissue constrained spherical deconvolution algorithm from the MRtrix3 package[86,87] (https://www.mrtrix.org/). After that, the structural connectivity weight between any two regions was given by the number of streamlines normalized by the mean length of streamlines and the mean surface area of the two regions. This normalization reduces bias towards long fibers during streamline reconstruction, as well as the bias from differences in region sizes.

**Functional time-series reconstruction**. Functional MRI data were corrected for gradient nonlinearity, head motion (using a rigid body transformation), and geometric distortions (using scan pairs with opposite phase encoding directions (R/L, L/R)[88]. BOLD time-series were then subjected to a high-pass filter (>2000s FWHM) to correct for scanner drifts, and to the ICA-FIX process to remove additional noise[89]. The data was parcellated in the same atlas used for structural networks.

**Time-resolved structure-function coupling**. To estimate region- and time-resolved structure-function coupling, we first constructed temporal co-fluctuation matrices. We started by calculating the element-wise product of the z-scored BOLD time-series between pairs of brain regions[14]. Region pairs with an activity on the same side of the baseline will have a positive co-fluctuation value, whereas two regions that fluctuate in opposite directions at the same time will have a negative co-fluctuation value (Fig. 1a). The average across time of these co-fluctuation matrices recovers the Pearson correlation coefficient that is often used to define functional connectivity.

To define region-specific structure-function coupling, we constructed a multilinear regression model to predict the co-fluctuation profile of a node $i$ from its geometric and structural connectivity profile to all other nodes $j \neq i$[23]. Predictors included Euclidean distance, shortest path length, and communicability. Euclidean distance was calculated between node centroids. Shortest path length refers to the shortest contiguous sequence of edges between 2 nodes. Communicability ($C_{ij}$) between two nodes $i$ and $j$ is defined as the weighted sum of all paths and walks between those nodes[23,49]. For a weighted adjacency matrix $A$, communicability is calculated as $C_{ij} = (\exp(D^{-1/2}AD^{-1/2}))_{ij}$, where $D = diag(\Sigma_{k=1}^{N} a_{ik})$ is the diagonal matrix of the generalized node degree matrix[50]. Shortest path length was implemented using Braincomm (https://github.com/fiuneuro/brainconn), a Python version of the Brain Connectivity Toolbox. Weighted communicability was implemented in netneurotools (https://github.com/netneurolab/netneurotools), an open-source Python package for network neuroscience. We used the minmax-normalized weighted structural connectivity matrix for each individual, and a negative log transformation was applied to the structural connectivity weights before calculating the shortest path length[51].

Concretely, for region $i$, subject $s$, time point $t$, we have,

$$cofluc_{s,t,i} = \beta_0 + \beta_1 dist_i + \beta_2 spl_{s,i} + \beta_3 cmc_{s,i} \qquad (1)$$

where $cofluc_{s,t,i}$ is the co-fluctuation profile, predicted by Euclidean distance $dist_i$, shortest path length $spl_{s,i}$ and weighted communicability $cmc_{s,i}$. The regression coefficients $\{\beta_0, \beta_1, \beta_2, \beta_3\}$ were estimated by ordinary least squares. Coupling was measured using adjusted R-squared $R_{i,t}^2$, a metric for goodness of fit. The regression was applied for individual profiles of brain regions, generating a cortical map of coupling values at each time point for each subject. We therefore define structure-function relationships as the goodness of fit for the linear regression model and, in keeping with previous literature, we refer to model fit as "coupling".

**Static and dynamic structure-function coupling**. The multilinear regression model, when applied without temporal expansion, generates one $R^2$ value per brain region, which we refer to here as *static* coupling[23]. By incorporating temporal co-fluctuation patterns, we obtained structure-function coupling measure $R_t^2$ per region as a frame-by-frame time-series, which we call *dynamic* coupling. To assess how dynamic coupling differs from static coupling, we frame the question as comparing a single observation (static) with a distribution (dynamic). We defined three summary statistics: (1) the probability of having a larger dynamic coupling value compared to the static coupling, (2) the bias, and (3) the variance of the dynamic coupling to reproduce the static values. Bias was used to evaluate how dynamic values deviate from the static value. It was calculated as the median of the difference between the dynamic coupling values and the static coupling. Small values of bias indicate that dynamic coupling values are close to the static coupling values, while large values indicate deviation. Variance was used to evaluate the extent of scattering of the dynamic values. It was calculated as the standard deviation of the distribution formed by the dynamic values. More specifically, we used the difference between the $84^{th}$ percentile and the $16^{th}$ percentile (±1 standard deviation from the mean under normality) to provide a more robust estimation of variability in case of possible outliers, extreme values, or skewed distributions. Thus a low variance value means that the distribution had low variability, and high variance value indicates the opposite.

**Cortical annotations**. Patterns of dynamic local structure-function coupling were contextualized relative to three common annotations: (1) 7 intrinsic functional networks as defined in ref. [40], 7 cytoarchitectonic classes described in ref. [42,90], and 10 functional hierarchy groups as defined in ref. [23], based on the principal functional gradient reported in ref. [41]. Collectively, these three partitions of the brain are thought to reflect multimodal hierarchies[91].

**Null models**. To assess correspondence between coupling maps and cortical annotations, we applied spatial autocorrelation-preserving permutation tests, termed "spin tests"[23,92,93]. In this model, the cortical surface is projected to a sphere using the coordinates of the vertex closest to the center of mass of each parcel. The sphere is then randomly rotated, generating surface maps with randomized topography, where each parcel has a reassigned value. The parcels corresponding to the medial wall were assigned to the closest rotated parcel[23,29,94]. The rotation was applied to one hemisphere and then mirrored to the other hemisphere. This corresponds to "Vázquez-Rodríguez" method described in ref. [93]. The method was chosen based on the benchmarking in ref. [93] because (a) it is was consistently most conservative method in the simulation and empirical analyses, (b) it was designed for parcellated data and did not have to discard permutations when parcels were rotated into the medial wall. We generated 10,000 spin permutations using netneurotools (https://github.com/netneurolab/netneurotools). Details of spatially-constrained null models in neuroimaging (https://github.com/netneurolab/markello_spatialnulls) were described in ref. [93].

**Predictor contributions**. Predictor contributions in the supplementary figure were calculated using dominance analysis[95,96]. The analysis estimates the relative importance of predictors by constructing all possible subsets of the predictor

variables and re-fitting the regression model for each combination. The "total dominance" statistic is adopted as a summary measure quantifying the contribution of each predictor to the overall goodness of fit. This method, among other procedures for interpreting multilinear regression models, can account for multi-collinearity and is sensitive to potential patterns in the model[97]. This paper used a re-implementation of the Dominance-Analysis (https://github.com/dominance-analysis/dominance-analysis) package in netneurotools (https://github.com/netneurolab/netneurotools).

## Data availability

The MRI data that support the findings of this study are available from the Human Connectome Project, https://www.humanconnectome.org/study/hcp-young-adult.

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

## Acknowledgements

We thank Golia Shafiei, Justine Hansen, Laura Suárez, Ross Markello, Vincent Bazinet, Louis-Philippe Robichaud and Filip Milisav for comments and suggestions on the manuscript. This research was undertaken thanks in part to funding from the Canada First Research Excellence Fund, awarded to McGill University for the Healthy Brains for Healthy Lives initiative. BM acknowledges support from the Natural Sciences and Engineering Research Council of Canada (NSERC Discovery Grant RGPIN #017-04265) and from the Canada Research Chairs Program.

## Author contributions

Conception of project: ZQL, BVR, BM. Analysis: ZQL, BVR. Writing – original draft: ZQL, BVR, BM. Writing – editing: RNS, BCB, RFB.

## Competing interests

The authors declare no competing interests.
