## [Peer Review File · Communications Biology]

Reviewers' comments:

Reviewer #1 (Remarks to the Author):

Dear authors

thanks for this manuscript, which contains an account on the statistical relationship between two families of measures, using tools recently developed by the different labs participating in this project.

The presentation is overall clear, and it appears to me that there are no flaws in the analysis pipeline.

I have some problems in grasping the biological significance and further insight provided by this work, beyond reporting some derived measures.

We have a time-varying measure (FC, here conjugated in time resolved co-fluctuations), and a family of static ones (SC-derived metrics). By definition this means that the relation between the two will fluctuate in time (and in space since there is spatial differentiation of the two). The fact that you refer to it as "coupling" or "tethering" confuses things rather than clarifying, since the nature of this relation is not expressed, beyond solving a linear regression.

1. "Here we show that structure-function coupling is dynamic and regionally heterogeneous": this is self-evident, and definitely not a discovery stemming from this work.

2. "We find that patterns of dynamic structure-function coupling are highly organized across the cortex". "Highly" is somewhat vague here, see below on the way you quantify this organization.

3. "These patterns reflect cortical hierarchies". What do you mean by "reflect"? I think you would have used the same word if there would have been a monotonically increasing or decreasing function, or the opposite "U-shaped" (we'll come back to this later) pattern. Same applies to the other use of "shaped" below in the abstract, referring to the connection lengths.

4. "The time-varying coupling of structural and functional connectivity points towards an informative feature of the brain that may reflect how cognitive functions are flexibly deployed and implemented". This is too general, and could apply to virtually any large scale neuroimaging study (being equally vague of course).

5. "As a result, structure-function coupling is also likely to fluctuate". Not "likely". It's the actual definition, in particular if by "coupling" you mean statistical similarity of a fixed quantity and a varying one.

6. Regression: why you choose these three measures (as opposed to the SC matrix itself for example)? Is there any particular generative model you have in mind that would connect these features to the fluctuations?

7. Figure 2.a "The coefficients span a wide distribution, encompassing both positive and negative values, suggesting that dynamic structure-function coupling provides a fundamentally different perspective on structure-function relationships.". Again, since structure is fixed, this is a characteristic of functional fluctuations themselves. As a matter of fact, if you compare the edge fluctuations to the static FC, running this simple code based on your edge centric demo

```
load example_data
ts = double(ts);
[T,N] = size(ts);
M = N*(N - 1)/2;
ets = fcn_edgets(ts);
FC=corr(ts);
ind=find(triu(ones(N),1));
CC=zeros(T,1);
for i=1:T
mat=zeros(N);
```

```

mat(ind)=ets(i,:);
mat=mat+mat';
CC(i)=corr(sum(FC,2),sum(mat,2));
end
histogram(CC,20)

```

you get values spanning roughly the same interval. And this is what you would expect comparing the fluctuations of a nonuniform time series around its mean, to the mean itself.

8. The medial views of panels 2d and 2f look very similar (identical by just looking at them). That could be I guess, but given that the lateral views are very different, maybe there's a problem

9. Figure 2: the distribution of the letters across the panels [a c d; b e f] is a bit arbitrary.

10. The use of a diverging colormap for purely positive (or purely negative) values is misleading (at least for me).

11. Comparison with Spearman rank coupling. You first state what you see as a weakness (missing out important dyadic relationships when there's no direct structural link), and one says of dair enough, but then confidently say that similar conclusions can be drawn (based on 0.22 correlation). This sounds a bit vague, yet reflects the true nature of the problem. There is correlation to some extent between structural and functional connectivity, we can describe it with a method or the other, but basically we cannot learn much on fundamental, true "coupling" from these measures, and one wonders whether we need yet another measure.

12. Hierarchical organization of coupling: what's your null model here? No organization at all? Or the monotonically decreasing relation found in Vázquez-Rodríguez et al. 2019? You write "the null hypothesis that $cv(R2)$ is greater than expected in intermediate positions of the unimodal-transmodal hierarchy". Unless I interpret in the wrong way, the null model should be that $cv(R2)$ is uniform across positions, right? Nothing is said about how many "intermediate" regions should reject the null, if you tested a specific U shape contrast for some reason, if the extremities are supposed to be "anchors" (another metaphor that does not help here), and what does this mean. To me it seems that you are just testing whether is there some specificity of individual position in the hierarchy (or ICN, or cytoarchitectonic class). And then you report the results as a "tendency" of (generically) "intermediate regions" to have values different than the others.

13. There's something off in the interpretation of figures 4b and 4c. "In particular, we find that regions intermediate in the unimodal-transmodal hierarchy, corresponding to the insular cortex, tend to have greater dynamic than static coupling (Fig. 4b). These regions also have the closest correspondence between dynamic and static coupling (Fig. 4c). This sentence is contradictory, but the reason is that in figure 4b the probability goes up to 0.5, not 1. This means that, unless the distribution is horribly skewed, probabilities closer to 0.5 mean by definition less bias. In this sense panels 4b and 4c make more sense (still with some redundant information).

14. Figure 4d. Is it actually variance you are looking at? Later you refer to it as the standard deviation of a trimmed distribution. If that's the case, maybe avoid calling it "variance" between quotation marks, since it can lead to confusion. "More specifically, we used the difference between the 84th percentile and the 16th percentile to avoid an underestimation for skewed distributions". Why are these values chosen? Is there a way for the reader to assess what this means in terms of the distribution?

15. The role of physiology, blood flow and respiration to start with, is never mentioned, while it always should for (resting state in particular) fMRI studies. This goes beyond saying that retroicor or any other regression was applied and that the methods applied here are robust, but rather lies in the fact that the BOLD signal itself, and thus any "coupling" with the structure, or with behavior, or whatever, passes by fluctuations in physiology, blood arrival time in different parts of the brain, etc.

Minor

- caption of Figure 1: "its" structural profile.

To conclude, I would summarize the message of this paper as "co-fluctuations of BOLD time series are moderately explained, sometimes with less error, sometimes with more error, by a mixture of three constant measures derived from structural connectivity, and the variability of the model fit is greater in some regions and lower in others". Thus in my opinion the concept of "coupling", the inverted U shape, and the interpretation in terms of what functions are usually associated with the two sets of regions with higher variability, are all not so justified, and confusing. Apart from the (very legitimate and laudable) motivation given by the intellectual curiosity of integrating our most recently developed methods in the field, I don't see an immediate biological interpretation, or novel insight. I don't think you will agree with me, maybe you can change my mind.

Thanks for listening.

Daniele Marinazzo

Reviewer #2 (Remarks to the Author):

This manuscript examines the coupling between structural connectivity and functional connectivity in the human brain a time-resolved manner. The proposed approach is simple: modelling the instantaneous functional connectivity between brain parcels (defined as the product of their fMRI signals) as a linear regression of three metrics: 1) the Euclidean distance between the parcels, 2) the shortest path length, and 3) communicability, where the two latter metrics are derived from the structural connectivity matrix based on diffusion MRI data and tractography. After fitting such a model in a time-dependent manner, the authors propose several summary statistics to investigate how the fitting of this dynamic functional-structural coupling model compares with a static one. The paper is very clear, and the methods and analyses performed are mostly based on previous works by the authors. To my understanding, from a methodological perspective, the novelty of the paper is the proposal of the linear regression model in a time-dependent manner based on the recently presented approach of edge-centric functional connectivity. To certain extent, the results of the paper are mainly observational. Although this does not mean that the paper and results are useful, the paper lacks predictions on the results. My main comments go in that direction, and in my opinion, they should be addressed before the paper is accepted.

1- In my opinion, the introduction must clearly state what the predictions or hypothesis are. For instance, did the authors expected that the insular cortex and frontal eye fields exhibit the largest dynamic coupling? Which would the predictions be based on a previous static models? Can we really assume that the model is linear (see Sarwar et al., Neuroimage 2021)

2- A recent manuscript by Novelli and Razi (<https://arxiv.org/abs/2106.10631>) has clearly demonstrated that edge-centric FC can be characterised as a function of the static node-centric FC. Therefore, what is the novelty in using edge-centric FC compared with previous models? In that sense, the authors propose to use Euclidean distance, which is node centric, along with shortest path length and weighted communicability, which to my understanding can be interpreted as edge-centric metrics. First of all, in my opinion, the paper should have explicitly defined these last two metrics. Although shortest path length is somehow a popular metric, I could not find any metric named "communicability" in the Python version of the Brain Connectivity Toolbox (<https://brainconn.readthedocs.io/en/latest/api.html>). More importantly, since the results depend on the goodness of fit for each node connectivity profile, and thus depend on the set of regressors, why the authors only used two complex network metrics? Why not also node strength, edge betweenness centrality, or even the structural connectivity matrix? In other words, can we assume that these three metrics are sufficient and the best model to describe the functional-structure coupling? Can we establish a hypothesis that these metrics are a good and sufficient and efficient model? Of course, adding more metrics means adding more degrees of freedom, which would result in more variance explained (i.e. larger R²), and one needs to consider adjusting for the number of regressors.

3- The mean R2 values over time and subjects are very, very low according to the range in Figure 2d. How do you explain this? On the other hand, in page 5, the authors state that "dynamic structure-function coupling is poorly correlated with multiple measures of structural and functional network embedding, including betweenness, clustering and degree". I find this statement a bit surprising because the correlation coefficients are not very low. To me, $r=-0.16$ for degree, $r=-0.197$ for mean edge length derived from SC are not really low. The correlation coefficients are even larger for functional connectivity metrics. What's the correlation with the metrics used in the proposed model? As noted previously, the R2 values of the proposed model are also very low. This should be clarify.

Minor comments:

- In Figures 2b and 5a (left), it is necessary to indicate what the diameter of each circle represents.
- In figure 3, it would be nice to use the same dark/light color in the boxes (similar to Figure S2) in order to indicate which ones are significant.
- In Figures 3c,d and e, it is not clear to me how the boxes computed? Do they represent variability across subjects, parcels, both?
- Why was the bias calculated as the median and not the mean of the difference? Would the results change?
- There seems to be a clear outlier in the plot of correlation betweenness centrality and coefficient of variation in Figure 5B.
- The definition of the variance should be clarify. Was it defined as the standard deviation or was it defined as the difference between two percentiles (so, something like a interquartile range)? If the former, then the paper should demonstrated that the distribution is Gaussian? If the latter, why were 16th and 84th percentiles chosen?
- The dominance analysis approach was employed to examine the relative importance of predictors. Could the authors explain the difference between this approach and the computation of F-statistics based on the RSS of the full model and reduced models? Would they give similar results?
- In my opinion, the title should be more specific. Brain structure can mean multiple things (anatomy, diffusion, cytoarchitecture, etc), whereas the paper only used diffusion based structural connectivity.
- The paper uses one of the null models studied in [49]. Due to the variability across all the models studied in [49] and the conclusions of that paper, my opinion is that the manuscript should explain why this null model was chosen for this particular application. In addition, it would be nice to refer to such null model with the same term as in [49].

Reviewer #1 (Remarks to the Author):

Dear authors

thanks for this manuscript, which contains an account on the statistical relationship between two families of measures, using tools recently developed by the different labs participating in this project.

The presentation is overall clear, and it appears to me that there are no flaws in the analysis pipeline.

I have some problems in grasping the biological significance and further insight provided by this work, beyond reporting some derived measures.

We have a time-varying measure (FC, here conjugated in time resolved co-fluctuations), and a family of static ones (SC-derived metrics). By definition this means that the relation between the two will fluctuate in time (and in space since there is spatial differentiation of the two). The fact that you refer to it as "coupling" or "tethering" confuses things rather than clarifying, since the nature of this relation is not expressed, beyond solving a linear regression.

We agree with the Reviewer - both terms are more suggestive than the underlying linear regression permits. We have removed the term "tethering" from the manuscript completely.

To facilitate comparison with other reports on the same topic, we have retained the term "coupling" because this is now a commonly-used term in the literature to describe structure-function relationships (Honey et al., 2009, Vazquez-Rodriguez et al., 2019, Baum et al., 2020, Suarez et al., 2020). We have, however, explicitly indicated in the revised manuscript that "coupling" is inferred from a linear regression and is thought to reflect structure-function relationships ("Methods" section, "Time-resolved structure-function coupling" subsection, paragraph #3)

"We therefore define structure-function relationships as the goodness of fit for the linear regression model and, in keeping with previous literature, we refer to model fit as "coupling"."

1. "Here we show that structure-function coupling is dynamic and regionally heterogeneous": this is self-evident, and definitely not a discovery stemming from this work.

We concur with the Reviewer that, if structural connectivity does not vary over time but functional connectivity does, then the correlation between the two should also vary over time. We have therefore changed the sentence in the Abstract to read:

"Here we study structure-function coupling from a dynamic perspective, and show that it is regionally heterogeneous."

However, we believe that the present manuscript makes two contributions to the literature on structure-function relationships. First, we develop a method to study and quantify how structure-function coupling varies over time. Previous studies typically assume that structure-function coupling is static and estimate it over the course of the whole scan. Second, we show that variation of structure-function coupling is not randomly or uniformly distributed across the brain, and instead follows the cortical functional hierarchy.

2. "We find that patterns of dynamic structure-function coupling are highly organized across the cortex". "Highly" is somewhat vague here, see below on the way you quantify this organization.

We agree with the Reviewer and have revised the sentence to be more specific in the Abstract:

"We find that patterns of dynamic structure-function coupling are region-specific."

The subsequent sentence in the Abstract hopefully provides additional clarity: "These patterns reflect the cortical hierarchy, with stable coupling in unimodal and transmodal cortex, and dynamic coupling in intermediate regions, particularly in insular cortex (salience network) and frontal eye fields (dorsal attention network).".

3. "These patterns reflect cortical hierarchies". What do you mean by "reflect"? I think you would have used the same word if there would have been a monotonically increasing or decreasing function, or the opposite "U-shaped" (we'll come back to this later) pattern. Same applies to the other use of "shaped" below in the abstract, referring to the connection lengths.

The original wording was meant to prime the reader that regional structure-function coupling is related to a region's position in the putative unimodal-transmodal cortical hierarchy. Given that this was in the Abstract, we wanted to give a gentle introduction to the concept first. We do, however, agree with the Reviewer that the wording could be more precise. We have revised the sentence to be more specific in the Abstract:

"These patterns display a U-shaped relationship with cortical hierarchies, ..."

We have also revised the following sentence about connection lengths:

"Finally, we show that the variability of a region's structure-function coupling is related to the distribution of its connection lengths."

4. "The time-varying coupling of structural and functional connectivity points towards an informative feature of the brain that may reflect how cognitive functions are flexibly deployed and implemented". This is too general, and could apply to virtually any large scale neuroimaging study (being equally vague of course).

Our original intent here was to point out that the insular cortex/salience network - which is typically associated with cognitive flexibility and is thought of as the interface between external perception and interoception - shows the greatest variability in structure-function coupling. We agree with the Reviewer and have revised the sentence to be more specific in the Abstract:

“Collectively, our findings provide a way to study structure-function relationships from a dynamic perspective.”

5. "As a result, structure-function coupling is also likely to fluctuate". Not "likely". It's the actual definition, in particular if by "coupling" you mean statistical similarity of a fixed quantity and a varying one.

We agree with the Reviewer and have remove the word “likely” (“Introduction” section, paragraph #2):

"As a result, structure-function coupling should fluctuate over multiple timescales."

6. Regression: why you choose these three measures (as opposed to the SC matrix itself for example)? Is there any particular generative model you have in mind that would connect these features to the fluctuations?

The structural connectivity matrix is sparse, so correlating the two matrices would exclude node pairs that do not share a direct structural connection. Structure-function coupling has been computed in this way before (e.g. Baum et al., 2019), and it is a method that we also adopted and showed in the supplement (Fig. 2b). However, we preferred to use three graph distance measures that would provide a value for every region pair, and showed results from this method in the main text.

We chose the three specific predictors because they reflect three canonical communication protocols. Namly, path length is a distance measure consistent with centralized, routing-like communication. Communicability is a distance measure consistent with decentralized, diffusion-like communication. Euclidean distance is a distance measure consistent with decentralized navigation. The measures we choose are in no way a complete profile of the structural attributes; however, they cover several well-studied and potentially biologically relevant computational models, and were robust in our previous study (Vazquez-Rodriguez et al., 2019).

To clarify why these specific measures were chosen, we have added the following passage to the revised manuscript (“Results” section, “Time-resolved structure-function coupling” subsection, paragraph #1):

“The regression model incorporates multiple computational models of cortical communication. Euclidean distance embodies the notion that proximal neurons

may exchange information more easily, and is consistent with navigation-like communication (Seguin et al., 2018). Shortest path length is a statistic that embodies centralized routing-like communication (Fornito et al., 2016), while communicability is a statistic that embodies decentralized diffusion-like communication.”

We have also added the following explanation that this not an exhaustive profile of all possible types of communication processes (“Results” section, “Time-resolved structure-function coupling” subsection, paragraph #1):

“Note however, that there exist multiple alternative statistics that measure the capacity of the network to transmit information, and the three chosen measures constitute a subset of that wider space (Avena-Koenigsberger et al., 2017).”

7. Figure 2.a "The coefficients span a wide distribution, encompassing both positive and negative values, suggesting that dynamic structure-function coupling provides a fundamentally different perspective on structure-function relationships.". Again, since structure is fixed, this is a characteristic of functional fluctuations themselves. As a matter of fact, if you compare the edge fluctuations to the static FC, running this simple code based on your edge centric demo

```
load example_data
ts = double(ts);
[T,N] = size(ts);
M = N*(N - 1)/2;
ets = fcn_edgets(ts);
FC=corr(ts);
ind=find(triu(ones(N),1));
CC=zeros(T,1);
for i=1:T
mat=zeros(N);
mat(ind)=ets(i,:);
mat=mat+mat';
CC(i)=corr(sum(FC,2),sum(mat,2));
end
histogram(CC,20)
```

you get values spanning roughly the same interval. And this is what you would expect comparing the fluctuations of a nonuniform time series around its mean, to the mean itself.

We understand the Reviewer’s concern that the functional time series itself exhibits fluctuations around the mean. We have substantially revised this section to appropriately reflect the point that, given static structural connectivity and dynamic functional connectivity, we expect dynamic

structure-function coupling. First, we have removed the part of the sentence that reads “suggesting that dynamic structure-function coupling provides a fundamentally different perspective on structure-function relationships.” We have also added text to emphasize the Reviewer’s point. The revised section reads (“Results” section, “Time-resolved structure-function coupling” subsection, paragraph #3):

“The coefficients span a wide distribution, encompassing both positive and negative values. This is consistent with the notion that temporal fluctuations in functional connectivity around static structural connectivity should result in dynamic structure-function coupling.”

At the same time, we note that fluctuations of structure-function coupling do not necessarily trivially recapitulate fluctuations in functional connectivity. The communication predictors are effectively nonlinear transforms of the underlying structural network, and the multilinear regression yields a final R^2 value which will also have a nonlinear relationship with structural connectivity. As a result, fluctuations in functional connectivity do not necessarily have to parallel fluctuations in structure-function coupling. Indeed, variability of moment-to-moment activity and functional connectivity in resting-state fMRI (computed using the same method) tends to separate unimodal and transmodal cortex (greatest fluctuations in default and control networks) (Esfahlani et al., 2020, Zalesky et al., 2014), whereas we find an entirely different pattern with the most prominent fluctuations in the insular cortex and frontal eye fields, corresponding to the salience network.

More generally, as we discussed in our reply to Point #1, the present work highlights the fact that, by their very nature and definition, the relationship between structural and functional connectivity is expected to be dynamic and should be studied from that perspective.

8. The medial views of panels 2d and 2f look very similar (identical by just looking at them). That could be I guess, but given that the lateral views are very different, maybe there's a problem

Thank you for pointing this out! We made an error and copied the medial view from 2d in 2f. We sincerely apologize for this overlooking and thank the Reviewer for pointing out this issue. We have corrected the panel in the revised manuscript (Revised Fig. 2 is shown below).

9. Figure 2: the distribution of the letters across the panels [a c d; b e f] is a bit arbitrary.

We agree with the Reviewer and have adjusted the panel lettering to follow a more coherent sequence (Revised Fig. 2 is shown above).

10. The use of a diverging colormap for purely positive (or purely negative) values is misleading (at least for me).

We agree with the Reviewer and have corrected the panel in the revised manuscript (Revised Fig. 2 is shown above), as well as other similar cases (Revised Figs. 3,5 is shown below.).

11. Comparison with Spearman rank coupling. You first state what you see as a weakness (missing out important dyadic relationships when there's no direct structural link), and one says ok fair enough, but then confidently says that similar conclusions can be drawn (based on 0.22 correlation). This sounds a bit vague, yet reflects the true nature of the problem. There is correlation to some extent between structural and functional connectivity, we can describe it with one method or the other, but basically we cannot learn much on fundamental, true "coupling" from these measures, and one wonders whether we need yet another measure.

We wanted to quantify the similarity between structural and functional connectivity over time. To our knowledge, two methods have been previously used in the literature: the multilinear model (Vazquez-Rodriguez et al., 2019) and the Spearman rank model (Baum et al., 2020). To facilitate comparisons with this literature, we decided to use one of these methods, rather than inventing a new one.

For the main text, we picked the method that we thought had more advantages and better represented the system we wanted to study (see our replies to point #6). For completeness, we also implemented the alternative method and presented it in the manuscript. As the Reviewer points out, the relationship between the two is not perfect but it is positive, providing some confidence that the results are not idiosyncratic to one method. In our opinion, picking an existing method, justifying its use, and also comparing it with an alternative method represents a comprehensive scientific approach to this particular empirical question.

At the same time, we agree with the Reviewer that the conclusion could be more nuanced. We have revised the sentence to read ("Results" section, "Time-resolved structure-function coupling" subsection, paragraph #3):

"Importantly, the two methods are positively correlated ($r=0.22$), suggesting that, while the two methods offer qualitatively similar perspectives on structure-function coupling, they are not perfectly correlated, potentially because one is sensitive to direct monosynaptic relationships while the other also takes into account polysynaptic relationships."

12. Hierarchical organization of coupling: what's your null model here? No organization at all? Or the monotonically decreasing relation found in Vázquez-Rodríguez et al. 2019? You write "the null hypothesis that $cv(R2)$ is greater than expected in intermediate positions of the unimodal-transmodal hierarchy". Unless I interpret in the wrong way, the null model should be that $cv(R2)$ is uniform across positions, right? Nothing is said about how many "intermediate" regions should reject the null, if you tested a specific U shape contrast for some reason, if the extremities are supposed to be "anchors" (another metaphor that does not help here), and what does this mean. To me it seems that you are just testing whether is there some specificity of individual position in the hierarchy (or ICN, or cytoarchitectonic class). And then you report the results as a "tendency" of (generically) "intermediate regions" to have values different than the others.

We apologize that the original description was not clear. In Fig. 3c,d,e, we test the null hypothesis that $cv(R2)$ is uniform across different partitions of the brain (resting state networks, hierarchical position, cytoarchitectonic class). For each partition, we implement the null hypothesis by randomly permuting class labels while preserving spatial autocorrelation. We do this using so-called “spin tests”, whereby the labels are projected to a sphere and randomly rotated. The method therefore allows us to assess whether $cv(R2)$ is enriched in any particular class, above and beyond what would be expected given the size, symmetry and spatial extent of that class.

We find consistent results across the three partitions: the intermediate classes have significantly greater $cv(R2)$ compared to the extreme classes, corresponding to the insular cortex/salience network.

Just a side note: most papers in the field that attempt to do similar analyses use no null model. The few that do, tend to use naive permutation tests that do not preserve spatial autocorrelation and are therefore unrealistically liberal. We hope that the Reviewer agrees that the test here is both appropriate for the experimental question and rigorous.

To clarify how the nulls were implemented, we have modified the following passage (“Results” section, “Hierarchical organization of dynamic structure-function coupling” subsection, paragraph #1):

“These observations are confirmed using spatial autocorrelation-preserving null models to test the null hypothesis that $cv(R2)$ is uniform across the brain. The tests reveal significantly greater $cv(R2)$ in intermediate positions of the unimodal-transmodal hierarchy (Fig. S2).”

We have also clarified the description of the test (“Methods” section, “Null models” subsection):

“To assess correspondence between coupling maps and cortical annotations, we applied spatial autocorrelation-preserving permutation tests, termed “spin tests” (Alexander-Bloch et al., 2018, Vazquez-Rodriguez et al., 2019, Markello and Misic, 2021). In this model, the cortical surface is projected to a sphere using the coordinates of the vertex closest to the center of mass of each parcel. The sphere is then randomly rotated, generating surface maps with randomized topography, where each parcel has a reassigned value. The parcels corresponding to the medial wall were assigned to the closest rotated parcel (Vazquez-Rodriguez et al., 2019, Shafiei et al., 2020, Hansen et al., 2021). The rotation was applied to one hemisphere and then mirrored to the other hemisphere. We generated 10000 spin permutations using *netneurotools* (<https://github.com/netneurolab/netneurotools>). Details of spatially-constrained null models in neuroimaging

(https://github.com/netneurolab/markello_spatialnulls) were described in (Markello and Misic, 2021).”

13. There's something off in the interpretation of figures 4b and 4c. "In particular, we find that regions intermediate in the unimodal-transmodal hierarchy, corresponding to the insular cortex, tend to have greater dynamic than static coupling (Fig. 4b). These regions also have the closest correspondence between dynamic and static coupling (Fig. 4c). This sentence is contradictory, but the reason is that in figure 4b the probability goes up to 0.5, not 1. This means that, unless the distribution is horribly skewed, probabilities closer to 0.5 mean by definition less bias. In this sense panels 4b and 4c make more sense (still with some redundant information).

We thank the Reviewer for the careful examination and interpretation. Indeed, in Fig. 4b-d we choose three statistics to characterize properties of the distribution from different angles. The panels are related to each other by definition, however, we think they also display unique and complementary views. The bias-variance metric pair (panels c and d) is a simple and intuitive way to describe a population and its relationship with a single point. Meanwhile, the probability in panel b alone provides more details about the actual distribution and makes more direct comparison with the static coupling value (“where does the static value sit in the distribution of dynamic ones”). Note that we did not assume the skewness of the distribution, and by describing the multifaceted nature of the relationship, reasonable conclusions can be drawn.

We understand the Reviewer’s concern and have revised the aforementioned sentence to be more accurate (“Results” section, “Relating static and dynamic structure-function coupling” subsection, paragraph #2):

“In particular, we find that regions intermediate in the unimodal-transmodal hierarchy, corresponding to the insular cortex, tend to have relatively greater dynamic than static coupling compared with other groups in the hierarchy (up to 0.5, Fig. 4b). These regions also have the closest correspondence between dynamic and static coupling (Fig. 4c) and the lowest dynamical variance around the static case (Fig. 4d).”

14. Figure 4d. Is it actually variance you are looking at? Later you refer to it as the standard deviation of a trimmed distribution. If that's the case, maybe avoid calling it "variance" between quotation marks, since it can lead to confusion. "More specifically, we used the difference between the 84th percentile and the 16th percentile to avoid an underestimation for skewed distributions". Why are these values chosen? Is there a way for the reader to assess what this means in terms of the distribution?

We used the difference between the 84th and 16th percentile, which corresponds to ± 1 standard deviation from the mean under normality. We chose this adjusted statistic because it provides a more robust estimation of variability in case of possible outliers, extreme values, or skewed distributions.

To make this point more clear, we have revised the following text in the revised manuscript ("Methods" section, "Static and dynamic structure-function coupling" subsection):

"More specifically, we used the difference between the 84th percentile and the 16th percentile (± 1 standard deviation from the mean under normality) to provide a more robust estimation of variability in case of possible outliers, extreme values, or skewed distributions."

15. The role of physiology, blood flow and respiration to start with, is never mentioned, while it always should for (resting state in particular) fMRI studies. This goes beyond saying that retroicor or any other regression was applied and that the methods applied here are robust, but rather lies in the fact that the BOLD signal itself, and thus any "coupling" with the structure, or with behavior, or whatever, passes by fluctuations in physiology, blood arrival time in different parts of the brain, etc.

We concur with the Reviewer that all studies looking at the BOLD signal should explicitly acknowledge these confounds. We have added the following passage to the revised manuscript ("Discussion" section, paragraph #7):

"Second, the dynamics of the BOLD signal itself are influenced by multiple physiological confounds, including blood flow and respiration (Colenbier et al., 2020, Tsvetanov et al., 2020). In the absence of concurrent measurements of cardiovascular and cerebrovascular factors, these results must be interpreted with caution."

Minor

- caption of Figure 1: "its" structural profile.

Thanks!

To conclude, I would summarize the message of this paper as "co-fluctuations of BOLD time series are moderately explained, sometimes with less error, sometimes with more error, by a mixture of three constant measures derived from structural connectivity, and the variability of the model fit is greater in some regions and lower in others". Thus in my opinion the concept of "coupling", the inverted U shape, and the interpretation in terms of what functions are usually associated with the two sets of regions with higher variability, are all not so justified, and confusing. Apart from the (very legitimate and laudable) motivation given by the intellectual curiosity of integrating our most recently developed methods in the field, I don't see an immediate biological interpretation, or novel insight. I don't think you will agree with me, maybe you can change my mind.

Thanks for listening.

Daniele Marinazzo

Thank you for the thorough review and for your constructive criticism. Hopefully our responses above have addressed your main concerns. We have significantly toned down claims that structure-function coupling is dynamic, because as the Reviewer correctly points out, this is expected. To summarize, we believe that the study addresses the question of structure-function relationships from a new perspective and, through a series of rigorous follow-ups, shows that fluctuations in structure-function coupling are highly region-specific.

Reviewer #2 (Remarks to the Author):

This manuscript examines the coupling between structural connectivity and functional connectivity in the human brain a time-resolved manner. The proposed approach is simple: modelling the instantaneous functional connectivity between brain parcels (defined as the product of their fMRI signals) as a linear regression of three metrics: 1) the Euclidean distance between the parcels, 2) the shortest path length, and 3) communicability, where the two latter metrics are derived from the structural connectivity matrix based on diffusion MRI data and tractography. After fitting such a model in a time-dependent manner, the authors propose several summary statistics to investigate how the fitting of this dynamic functional-structural coupling model compares with a static one. The paper is very clear, and the methods and analyses performed are mostly based on previous works by the authors. To my understanding, from a methodological perspective, the novelty of the paper is the proposal of the linear regression model in a time-dependent manner based on the recently presented approach of edge-centric functional connectivity. To certain extent, the results of the paper are mainly observational. Although this does not mean that the paper and results are useful, the paper lacks predictions on the results. My main comments go in that direction, and in my opinion, they should be addressed before the paper is accepted.

1- In my opinion, the introduction must clearly state what the predictions or hypothesis are. For instance, did the authors expected that the insular cortex and frontal eye fields exhibit the largest dynamic coupling? Which would the predictions be based on a previous static models? Can we really assume that the model is linear (see Sarwar et al., Neuroimage 2021)

We agree with the Reviewer that the manuscript could be more informative for readers if predictions are clearly laid out. The majority of the previous literature on structure-function coupling assumes that the same relationship exists for all brain regions (Suarez et al., 2020), which would correspond to uniform dynamic structure-function coupling across the brain. Prior to conducting the study, we thought that two alternative hypotheses would be more likely:

1. Greater variability in structure-function coupling in transmodal cortex. The reason for this is two-fold. First, transmodal cortex engages in a variety of polysensory, associative

functions that may necessitate fluid re-routing of information and manifest as variable structure-function coupling. Second, we and others have previously shown that static structure-function coupling is lower in transmodal cortex compared to unimodal cortex (Vazquez-Rodriguez et al., 2019, Baum et al., 2020, Suarez et al., 2020, Valk et al., 2021). A plausible explanation could be that greater variability in time-dependent structure-function coupling ultimately averages out and appears as lower static structure-function coupling. We thought this was the most likely possibility.

2. Greater variability in insular cortex and the salience network. As we discuss in the manuscript, this hypothesis was largely driven by previous literature showing that insular cortex has diverse microstructure and connectional fingerprints, and therefore engages in diverse functions (Uddin, 2021).

We have added the following passage to the manuscript:

“We considered two alternative possibilities. One possibility is that structure-function coupling is greater in transmodal cortex. Several recent studies have shown that static structure-function coupling is lower in transmodal cortex compared to unimodal cortex (Vazquez-Rodriguez et al., 2019, Baum et al., 2020, Suarez et al., 2020, Valk et al., 2021). Given that transmodal cortex engages in multiple polysensory functions and functional relationships, a plausible explanation could be that greater variability in time-dependent structure-function coupling ultimately averages out and appears as lower static structure-function coupling. Another possibility is that structure-function coupling is greatest in the salience network and in insular cortex. Numerous evidence points to diverse cytoarchitecture and connectional fingerprints in insular cortex (Allen, 2020, Uddin, 2015). By participating in a diverse set of connections with multiple brain regions, the insula is thought to dynamically engage in multiple cognitive systems (Ghaziri et al., 2017, Kurth et al., 2010, Tian and Zalesky, 2018).”

2- A recent manuscript by Novelli and Razi (<https://arxiv.org/abs/2106.10631>) has clearly demonstrated that edge-centric FC can be characterised as a function of the static node-centric FC. Therefore, what is the novelty in using edge-centric FC compared with previous models? In that sense, the authors propose to use Euclidean distance, which is node centric, along with shortest path length and weighted communicability, which to my understanding can be interpreted as edge-centric metrics. First of all, in my opinion, the paper should have explicitly defined these last two metrics. Although shortest path length is somehow a popular metric, I could not find any metric named "communicability" in the Python version of the Brain Connectivity Toolbox (<https://brainconn.readthedocs.io/en/latest/api.html>). More importantly, since the results depend on the goodness of fit for each node connectivity profile, and thus depend on the set of regressors, why the authors only used two complex network metrics? Why not also node strength, edge betweenness centrality, or even the structural connectivity matrix? In other words, can we assume that these three metrics

are sufficient and the best model to describe the functional-structure coupling? Can we establish a hypothesis that these metrics are a good and sufficient and efficient model? Of course, adding more metrics means adding more degrees of freedom, which would result in more variance explained (i.e. larger R²), and one needs to consider adjusting for the number of regressors.

We agree with the Reviewer that several aspects could have been made more clear and we address the Reviewer's concerns as follows:

Re: co-fluctuation estimation

We wanted to quantify the similarity between structural and functional connectivity over time. To our knowledge, the functional edge co-fluctuation method is a simple and robust way to provide moment-to-moment dynamics of FC. We apply this method to study and quantify how structure-function coupling varies over time, while previous studies typically assume that structure-function coupling is static and estimate it over the course of the whole scan. Specifically, we focused on characterizing variation of structure-function coupling and found that it is not randomly nor uniformly distributed across the brain, and instead follows the cortical functional hierarchy.

Novelli and Razi (2021) presented a null-model-based analysis of the edge-centric FC, showing that it can produce some key findings in (Esfahlani et al., 2020) and (Faskowitz et al. 2020). However, we did not use edge-centric FC in this analysis, and we did not focus on its relationship with the static FC. Instead, we calculated only the edge co-fluctuation time series, studied its moment-to-moment dynamic properties, and its relationship to structural connectivity.

Interestingly, the Novelli and Razi paper specifically discussed the possibility and potential use of these “temporally-unfolded (or point-wise) dependence measures” to study structure-function coupling. They state that “the fact that such a null model is able to replicate the edge-centric features could be an indication that the temporal structure of the edge time series has not been fully exploited”. In our manuscript, we are looking precisely at the distribution properties displayed by structure-function coupling. The Novelli and Razi paper also shows that their null model can reproduce edge-centric FC when “only accounting for the observed static spatial correlations and not the temporal ones”. In our paper, we used a recent and more conservative spatial autocorrelation-preserving null model. We think this statistical test is more appropriate than widely-used cortically uniform permutation tests, and thus can address this critical claim and capture the useful structure-function coupling properties as predicted by Novelli and Razi.

We have added the following passage to the revised manuscript to acknowledge this point (“Discussion” section, paragraph #7):

“The statistical properties of the underlying dynamic processes behind moment-to-moment functional dynamics of the human brain has been an area of active research for years (Tagliazucchi et al., 2012, Tagliazucchi et al., 2016,

Petridou et al., 2013, Wu et al., 2013, Liu and Duyn, 2013, Karahanoğlu and Van De Ville, 2015), and the applicability of these methods to studying structure-function relationships is increasingly recognized (Novelli and Razi, 2021, Pope et al., 2021).”

Tagliazucchi, E., Balenzuela, P., Fraiman, D., & Chialvo, D. (2012). Criticality in Large-Scale Brain fMRI Dynamics Unveiled by a Novel Point Process Analysis. *Frontiers in Physiology*, 3, 15. <https://doi.org/10.3389/fphys.2012.00015>

Tagliazucchi, E., Siniatchkin, M., Laufs, H., & Chialvo, D. R. (2016). The Voxel-Wise Functional Connectome Can Be Efficiently Derived from Co-activations in a Sparse Spatio-Temporal Point-Process. *Frontiers in Neuroscience*, 10, 381. <https://doi.org/10.3389/fnins.2016.00381>

Petridou, N., Gaudes, C. C., Dryden, I. L., Francis, S. T., & Gowland, P. A. (2013). Periods of rest in fMRI contain individual spontaneous events which are related to slowly fluctuating spontaneous activity. *Human Brain Mapping*, 34(6), 1319–1329.

<https://doi.org/10.1002/hbm.21513>

Wu, G.-R., Liao, W., Stramaglia, S., Ding, J.-R., Chen, H., & Marinazzo, D. (2013). A blind deconvolution approach to recover effective connectivity brain networks from resting state fMRI data. *Medical Image Analysis*, 17(3), 365–374. <https://doi.org/10.1016/j.media.2013.01.003>

Liu, X., & Duyn, J. H. (2013). Time-varying functional network information extracted from brief instances of spontaneous brain activity. *Proceedings of the National Academy of Sciences*, 110(11), 4392–4397. <https://doi.org/10.1073/pnas.1216856110>

Karahanoğlu, F. I., & Van De Ville, D. (2015). Transient brain activity disentangles fMRI resting-state dynamics in terms of spatially and temporally overlapping networks. *Nature Communications*, 6(1), 7751. <https://doi.org/10.1038/ncomms8751>

Re: network metrics

We have revised the following text to define the predictors and the implementations (“Methods” section, “Time-resolved structure-function coupling” subsection, paragraph #2):

“Predictors included Euclidean distance, shortest path length, and communicability. Euclidean distance was calculated between node centroids. Shortest path length refers to the shortest contiguous sequence of edges between 2 nodes. Communicability (C_{ij}) between 2 nodes i and j is defined as the weighted sum of all paths and walks between those nodes (Estrada and Hatano, 2008, Vazquez-Rodriguez et al., 2019). For a weighted adjacency matrix A , communicability is calculated as $C_{ij} = (\exp(D^{-1/2}AD^{-1/2}))_{ij}$, where $D = \text{diag}(\sum_{k=1}^N a_{ik})$ is the diagonal matrix of the generalized node degree matrix (Crofts and Higham, 2009). Shortest path length was implemented using Brainconn (<https://github.com/fiuneuro/brainconn>), a Python version of the Brain Connectivity Toolbox. Weighted communicability was implemented in *netneurotools* (<https://github.com/netneurolab/netneurotools>), an open-source Python package for network neuroscience. We used the minimax-normalized

weighted structural connectivity matrix for each individual, and a negative log transformation was applied to the structural connectivity weights before calculating the shortest path length (Avena-Koenigsberger et al., 2017).”

Re: predictor choice

The structural connectivity matrix is sparse, so correlating the two matrices would exclude node pairs that do not share a direct structural connection. Structure-function coupling has been computed in this way before (e.g. Baum et al., 2019), and it is a method that we also adopted and showed in the supplement (Fig. 2b). However, we preferred to use three graph distance measures that would provide a value for every region pair, and showed results from this method in the main text.

We chose the three specific predictors because they reflect three canonical communication protocols. Namely, path length is a distance measure consistent with centralized, routing-like communication. Communicability is a distance measure consistent with decentralized, diffusion-like communication. Euclidean distance is a distance measure consistent with decentralized navigation. The measures we choose are not a complete profile of the structural attributes; however, they cover multiple canonical and biologically important computational models (Avena-Koenigsberger et al., 2017), and proved robust in our previous study (Vazquez-Rodriguez et al., 2019).

To clarify why these specific measures were chosen, we have added the following passage to the revised manuscript (“Results” section, “Time-resolved structure-function coupling” subsection, paragraph #1):

“The regression model incorporates multiple computational models of cortical communication. Euclidean distance embodies the notion that proximal neurons may exchange information more easily, and is consistent with navigation-like communication (Seguin et al., 2018). Shortest path length is a statistic that embodies centralized routing-like communication (Fornito et al., 2016), while communicability is a statistic that embodies decentralized diffusion-like communication.”

We have also added the following explanation that this not an exhaustive profile of all possible types of communication processes (“Results” section, “Time-resolved structure-function coupling” subsection, paragraph #1):

“Note however, that there exist multiple alternative statistics that measure the capacity of the network to transmit information, and the three chosen measures constitute a subset of that wider space (Avena-Koenigsberger et al., 2017).”

3- The mean R2 values over time and subjects are very, very low according to the range in Figure 2d. How do you explain this? On the other hand, in page 5, the authors state

that "dynamic structure-function coupling is poorly correlated with multiple measures of structural and functional network embedding, including betweenness, clustering and degree". I find this statement a bit surprising because the correlation coefficients are not very low. To me, $r=-0.16$ for degree, $r=-0.197$ for mean edge length derived from SC are not really low. The correlation coefficients are even larger for functional connectivity metrics. What's the correlation with the metrics used in the proposed model? As noted previously, the R^2 values of the proposed model are also very low. This should be clarify.

Re: low R^2 values

We agree with the reviewer that they have relatively low values, and even lower when averaged over time and subjects. We argue that this is expected and our results still provide solid conclusions for the following reason:

R^2 values are expected to be generally lower when calculated from individuals and moment-to-moment FC, in contrast to group-derived connectomes or FC taken from the full time course. The former case carries out regression with substantially less information than the latter, but acts at a finer temporal resolution and reveals more individual differences. When taking the average across time and subjects, in this case, higher R^2 values may be more sparse both temporally and nodally (Esfahlani et al., 2020), causing the mean R^2 values to be even lower. However, they are still meaningful through the variability statistics. For the exact reason, we adopted the coefficient of variation as a robust variability statistic that takes into account absolute R^2 values.

Re: correlations of R^2 with network metrics

We understand the Reviewer's concern about correlation with graph metrics. Here we included the correlations with several noteworthy graph metrics mainly to provide a more comprehensive view at the roles of R^2 values in the connectome. We have revised the manuscript to remove adjectives like high and low, so as not to over-interpret these findings.

Minor comments:

We thank the Reviewer for detailed comments, and will address them below.

- In Figures 2b and 5a (left), it is necessary to indicate what the diameter of each circle represents.

We clarify as follows and have added descriptions to the corresponding figure captions:
In Fig. 2b, the diameter of the circles are proportional to scatter point density (also shown in a sequential colormap). In Fig. 5a, the diameter of the circles are proportional to the standard deviation (also shown in a sequential colormap).

- In figure 3, it would be nice to use the same dark/light color in the boxes (similar to Figure S2) in order to indicate which ones are significant.

In Figures 3c,d and e, it is not clear to me how the boxes computed? Do they represent variability across subjects, parcels, both?

We used different colors mainly to label the different cortical annotations: principal functional gradient groups, intrinsic functional networks, and cytoarchitectonic classes. We think this may help the readers navigate between the brain areas and their boxes.

We apologize for the confusion. The values in Fig. 3c-e are using the mean cv(R2) values in Fig. 3a. They represent the temporal variability for each node (parcel) on the cortex, and are then averaged across subjects to show the general pattern.

- Why was the bias calculated as the median and not the mean of the difference? Would the results change?

When calculating the bias and variance metrics, we hope to minimize the influence of outliers. We are using median as a more robust statistic to represent scattering of the values (Leys et al., 2013).

- There seems to be a clear outlier in the plot of correlation betweenness centrality and coefficient of variation in Figure 5B.

We have run comparison between original Pearson correlation and two other robust estimation methods as shown below:

metric	r_pearson	r_bicor	r_percbend
betweenness centrality	0.0095	0.0031	-0.0152
clustering coefficient	-0.0287	-0.0349	-0.0359
degree	-0.1667	-0.1722	-0.1605
mean edge length	-0.1977	-0.1987	-0.1936
strength	0.2350	0.1956	0.1803
participation coefficient	0.3914	0.1647	0.2980

r_pearson represents the original Pearson correlation coefficient. r_bicor represents biweight midcorrelation, and r_percbend represents percentage bend correlation, both are robust methods that will down-weight outliers. The statistics were implemented in Python package Pingouin (Vallat, 2018).

We have added information from both to the revised manuscript (Figure 5 caption):

“To account for the possible effects of outliers, we also estimated these relationships using the biweight midcorrelation ($r =$

0.0031;-0.0349;-0.1722;-0.1987;0.1956;0.1647\$) and percentage bend correlation ($r = -0.0152;-0.0359;-0.1605;-0.1936;0.1803;0.2980$ \$).”

- The definition of the variance should be clarify. Was it defined as the standard deviation or was it defined as the difference between two percentiles (so, something like a interquartile range)? If the former, then the paper should demonstrated that the distribution is Gaussian? If the latter, why were 16th and 84th percentiles chosen?

We used the difference between the 84th and 16th percentile, which corresponds to ± 1 standard deviation from the mean under normality. We chose this adjusted statistic because it provides a more robust estimation of variability in case of possible outliers, extreme values, or skewed distributions.

To make this point more clear, we have revised the following text in the revised manuscript (“Methods” section, “Static and dynamic structure-function coupling” subsection):

“More specifically, we used the difference between the 84th percentile and the 16th percentile (± 1 standard deviation from the mean under normality) to provide a more robust estimation of variability in case of possible outliers, extreme values, or skewed distributions.”

- The dominance analysis approach was employed to examine the relative importance of predictors. Could the authors explain the difference between this approach and the computation of F-statistics based on the RSS of the full model and reduced models? Would they give similar results?

We were trying to provide a way to compare the contribution of the three predictors both within and possibly between regression models. While calculating F-test or R^2 difference in a leave-one-predictor-out setting is an effective way to assess contribution of an individual predictor in a single regression, they may suffer from multicollinearity between predictors, and also cannot be effectively compared between regressions. We adopted dominance analysis to quantitatively support the interpretation of multiple regression models (Kraha et al., 2012). It takes into consideration all possible subsets of the predictor variables and outputs individual dominance value that sums to R^2 of the full model. The details of the model specification can be found in (Budescu, 1993) and (Azen and Budescu, 2003), and in the “Methods” section, “Dominance Analysis” subsection of the manuscript. Although this is not the only possible statistical method for this purpose, it is suitable and applicable in the present analysis..

We have added the following section to the Methods (“Methods” section, “Predictor contributions” subsection):

“Predictor contributions in the supplementary figure were calculated using dominance analysis (Budescu, 1993, Azen and Budescu, 2003). The analysis estimates the relative importance of predictors by constructing all possible

subsets of the predictor variables and re-fitting the regression model for each combination. The “total dominance” statistic is adopted as a summary measure quantifying the contribution of each predictor to the overall goodness of fit. This method, among other procedures for interpreting multilinear regression models, can account for multicollinearity and is sensitive to potential patterns in the model (Kraha et al., 2012). This paper used a re-implementation of the Dominance-Analysis (<https://github.com/dominance-analysis/dominance-analysis>) package in *netneurotools* (<https://github.com/netneurolab/netneurotools>).

- In my opinion, the title should be more specific. Brain structure can mean multiple things (anatomy, diffusion, cytoarchitecture, etc), whereas the paper only used diffusion based structural connectivity.

We understand the Reviewer’s concern. We used the term as a reference to a series of previous papers on the topic (Honey et al., 2009, Vazquez-Rodriguez et al., 2019, Baum et al., 2020, Suarez et al., 2020). We have also defined structure as structural connectivity in the Abstract and at the beginning of the Results section.

- The paper uses one of the null models studied in [49]. Due to the variability across all the models studied in [49] and the conclusions of that paper, my opinion is that the manuscript should explain why this null model was chosen for this particular application. In addition, it would be nice to refer to such null model with the same term as in [49].

We have revised the text to make this more clear (“Methods” section, “Null models” subsection):

“This corresponds to “Vázquez-Rodríguez” method described in (Markello and Misić, 2021). The method was chosen based on the benchmarking in (Markello and Misić, 2021) because (a) it is was consistently most conservative method in the simulation and empirical analyses, (b) it was designed for parcellated data and did not have to discard permutations when parcels were rotated into the medial wall.”

Reviewers' comments:

Reviewer #1 (Remarks to the Author):

Dear colleagues

thanks for taking the time to read and reply. I appreciate the fact that you smoothed some claims and terminology. Still I have some doubts that you can maybe clarify:

1.

"The coefficients span a wide distribution, encompassing both positive and negative values. This is consistent with the notion that temporal fluctuations in functional connectivity around static structural connectivity should result in dynamic structure-function coupling."

These is a truism, not just mathematically (mathematical truisms can still be interesting), but semantically.

2. You still don't test for the presence of a U shaped relationship, yet you keep mentioning it.

See for example

Lind, J. T., & Mehlum, H. (2010). With or without U? The appropriate test for a U-shaped relationship. *Oxford bulletin of economics and statistics*, 72(1), 109-118.

or

Simonsohn, U. (2018). Two lines: A valid alternative to the invalid testing of U-shaped relationships with quadratic regressions. *Advances in Methods and Practices in Psychological Science*, 1(4), 538-555.

This latter comes with R code.

On the other hand you just check which partitions have a $cv(R^2)$ compatible with the shuffling or not. If you compare them on the basis of one rejecting the null and the other not, you fall into this mistake

Nieuwenhuis, S., Forstmann, B. U., & Wagenmakers, E. J. (2011). Erroneous analyses of interactions in neuroscience: a problem of significance. *Nature neuroscience*, 14(9), 1105-1107.

if on the other hand you just focus on those for which the value is outside the distribution, the shape could be several things other than a U.

Related, what's the fit of the right panel of figure 5a? How is it chosen or justified?

So the main problem is still that the interpretation of this higher variability in some partitions, in terms of cognitive functions, synaptic weights, and the like, appears too extreme in light of the weak evidence for the pattern/tendency that you observe.

Minor: "tethering" is still mentioned in a figure, while it's missing from the main text.

Reviewer #2 (Remarks to the Author):

I would like to thank the authors for their responses to my comments on the previous version of the manuscript. In my opinion, the paper is clearer and has now more relevant information. Having said that, in my opinion, the following points should be clarified further in order to accept the paper:

1- Regarding the predictions or hypothesis of the paper, I find a bit contentious the rationale for the second hypothesis about greater variability in insular cortex and salience network. Why only the insula and salience network? To me, they are also transmodal regions or networks, respectively, but more importantly why not other regions that also "participate in a diverse set of connections with multiple brain regions"? For example, precuneus and posterior cingulate cortex are also hubs of the human brain that diversively connect with multiple brain regions (van den Heuvel and Sporns, 2013; Tomasi and Volkow, 2011). Consequently, the formulation of this hypothesis should be motivated further.

2- Regarding the possible effect of outliers on the metrics, the table with the robust estimation methods seems to indicate that degree, mean edge length and strength are the only metrics that do not change considerably. This should be discussed in the main text of the paper.

3- The authors used the difference between the 84th and 16th percentile, which corresponds to ± 1 standard deviation from the mean under normality. However, the paper does not demonstrate whether the data follows a normal distribution.

4- The title is still very general and, in my opinion, justifying this choice in terms of a series of previous papers on the topic (which are also very general in my opinion) can be questionable. However, I understand your claim and I leave to the editors decide whether the title should be changed to describe the content of the work better.

Reviewer #1 (Remarks to the Author):

Dear colleagues

thanks for taking the time to read and reply. I appreciate the fact that you smoothed some claims and terminology. Still I have some doubts that you can maybe clarify:

1. “The coefficients span a wide distribution, encompassing both positive and negative values. This is consistent with the notion that temporal fluctuations in functional connectivity around static structural connectivity should result in dynamic structure-function coupling.”

These is a truism, not just mathematically (mathematical truisms can still be interesting), but semantically.

We agree, and have changed the passage to read:

“The coefficients span a wide distribution, encompassing both positive and negative values. A distribution of coefficients is mathematically expected given that the method is measuring the relationship between dynamic functional connectivity and static structural connectivity. In the present report, we further analyze how dynamic functional connectivity around a static structural connectivity reference yields fluctuations in structure-function correlations, and we map these fluctuations to the cortical hierarchy.”

It is worth noting that recent papers have specifically discussed the possibility and potential use of these “temporally-unfolded (or point-wise) dependence measures” to study structure-function relationship (e.g. Novelli & Razi, 2021; Ladwig et al., 2022). For instance, Novelli and Razi (2021) state that “the fact that such a null model is able to replicate the edge-centric features could be an indication that the temporal structure of the edge time series has not been fully exploited”. In our manuscript, we are looking precisely at the distribution properties displayed by structure-function coupling and contextualizing these patterns with respect to multiple features of brain anatomy.

Re: biological significance.

More broadly, our results echo predictions that the static anatomical connectivity can support a wide repertoire of functional configurations (Harris-Warrick & Marder, 1991). For instance, Getting and Deakin (1985) defined the concept of a “polymorphic network” as the moment-to-moment reconfiguration of a neural circuit to produce multiple different motor patterns. The hypothesis was directly based on electrophysiological recordings in the mollusc *Tritonia*, whereby the ganglion network can produce both a defensive withdrawal reflex and escape swimming. Follow-up work showed similar phenomena where anatomically circumscribed circuits can reconfigure due to normodulation by peptides and monoamines to

produce drastically different rhythmic motor patterns (Flamm & Harris-Warrick 1986a,b, Hooper & Marder 1987, Turrigiano & Selverston 1989). Ultimately, we believe that these findings capture a similar, biologically important phenomenon at the macroscale level, and will be interesting for the field (Suarez et al., 2020).

Harris-Warrick, R. M., & Marder, E. (1991). Modulation of neural networks for behavior. *Annual Review of Neuroscience*, 14, 39–57. <https://doi.org/10.1146/annurev.ne.14.030191.000351>

Getting, P. A., & Dikin, M. S. (1985). Tritonia Swimming. In A. I. Selverston (Ed.), *Model Neural Networks and Behavior* (pp. 3–20). Springer US. https://doi.org/10.1007/978-1-4757-5858-0_1

Flamm, R. E., & Harris-Warrick, R. M. (1986). Aminergic modulation in lobster stomatogastric ganglion. I. Effects on motor pattern and activity of neurons within the pyloric circuit. *Journal of neurophysiology*, 55(5), 847-865.

Flamm, R. E., & Harris-Warrick, R. M. (1986). Aminergic modulation in lobster stomatogastric ganglion. II. Target neurons of dopamine, octopamine, and serotonin within the pyloric circuit. *Journal of Neurophysiology*, 55(5), 866-881.

Hooper, S. L., & Marder, E. (1987). Modulation of the lobster pyloric rhythm by the peptide proctolin. *Journal of Neuroscience*, 7(7), 2097-2112.

Turrigiano, G. G., & Selverston, A. I. (1989). Cholecystinin-like peptide is a modulator of a crustacean central pattern generator. *Journal of Neuroscience*, 9(7), 2486-2501.

Suárez, L. E., Markello, R. D., Betzel, R. F., & Misic, B. (2020). Linking Structure and Function in Macroscale Brain Networks. *Trends in Cognitive Sciences*, 24(4), 302–315. <https://doi.org/10.1016/j.tics.2020.01.008>

2. You still don't test for the presence of a U shaped relationship, yet you keep mentioning it.

See for example

Lind, J. T., & Mehlum, H. (2010). With or without U? The appropriate test for a U-shaped relationship. *Oxford bulletin of economics and statistics*, 72(1), 109-118.

or

Simonsohn, U. (2018). Two lines: A valid alternative to the invalid testing of U-shaped relationships with quadratic regressions. *Advances in Methods and Practices in Psychological Science*, 1(4), 538-555.

This latter comes with R code.

On the other hand you just check which partitions have a $cv(R^2)$ compatible with the shuffling or not. If you compare them on the basis of one rejecting the null and the other not, you fall into this mistake

Nieuwenhuis, S., Forstmann, B. U., & Wagenmakers, E. J. (2011). Erroneous analyses of interactions in neuroscience: a problem of significance. *Nature neuroscience*, 14(9), 1105-1107.

if on the other hand you just focus on those for which the value is outside the distribution, the shape could be several things other than a U.

Related, what's the fit of the right panel of figure 5a? How is it chosen or justified?

Thanks - removed!

So the main problem is still that the interpretation of this higher variability in some partitions, in terms of cognitive functions, synaptic weights, and the like, appears too extreme in light of the weak evidence for the pattern/tendency that you observe.

We concur with the Reviewer. We did not explicitly test for a U-shape relationship, and this is not an important part of the results. We have therefore removed all descriptions of such a relationship.

Abstract. The original sentence was:

“These patterns display a U-shaped relationship with cortical hierarchies, with stable coupling in unimodal and transmodal cortex, and dynamic coupling in intermediate regions...”

We have changed the sentence to read:

“We observe stable coupling in unimodal and transmodal cortex, and dynamic coupling in intermediate regions...”

Results, “Hierarchical organization of dynamic structure-function coupling” subsection, paragraph #1. We have completely removed the following sentence:

~~“Altogether, the results show that structure-function coupling has an inverted U-shape relationship with cortical hierarchies.”~~

Results, “Relating static and dynamic structure-function coupling” subsection, paragraph #2. We have completely removed the following sentence:

~~“We again observe a U-shape relationship with cortical hierarchies.”~~

Results, “Spatial and topological determinants of dynamic structure-function coupling” subsection, paragraph #1. The original sentence reads:

“These regional differences follow an inverted U-shape relationship with dynamic structure-function coupling, such that areas with ...”

We have modified the sentence to read:

“We find that areas with ...”

Discussion, paragraph #2. The original sentence reads:

“Namely, we find that variability in coupling follows an inverted-U shape relative to the unimodal-transmodal hierarchy: the extremes of the hierarchy display more stable structure-function coupling, while regions intermediate in the hierarchy display more sizable fluctuations.”

We have modified the sentence to read:

“Namely, we find that the extremes of the putative unimodal-transmodal hierarchy display more stable structure-function coupling, while regions intermediate in the hierarchy display more sizable fluctuations.”

Discussion, paragraph #2. The original sentence reads:

“Namely, we also find an inverted-U shape relationship between connectivity distance and variability in structure-function coupling, such that regions with very short or very long connectivity distance tend to display stable coupling,...”

We have modified the sentence to read:

“Namely, we also find that regions with very short or very long connectivity distance tend to display stable coupling,...”

Nevertheless, we found the Reviewer’s suggestion to explicitly test for a U-shaped relationship inspiring, and we performed such an analysis for our own curiosity. We applied the suggested two-lines method from Simonsohn (2018) to the result in Figure 3c (<http://webstimate.org/twolines/>).

Confirming our intuition, the slopes are significantly positive before the inflection and significantly negative after the inflection. Just for fun, we further sought to test whether this would hold when including the spin null models (Fig S2a). To do so, we z-scored the R^2 value for each node relative to a population of 10,000 spin values. We then re-applied the “two line” method:

Again, we observe a similar result, with a positive slope before the inflection and a negative slope after.

Minor: "tethering" is still mentioned in a figure, while it's missing from the main text.

Sorry about that! We have removed the phrase from Figure 1c and Figure 5c.

Reviewer #2 (Remarks to the Author):

I would like to thank the authors for their responses to my comments on the previous version of the manuscript. In my opinion, the paper is clearer and has now more relevant

information. Having said that, in my opinion, the following points should be clarified further in order to accept the paper:

1- Regarding the predictions or hypothesis of the paper, I find a bit contentious the rationale for the second hypothesis about greater variability in insular cortex and salience network. Why only the insula and salience network? To me, they are also transmodal regions or networks, respectively, but more importantly why not other regions that also "participate in a diverse set of connections with multiple brain regions"? For example, precuneus and posterior cingulate cortex are also hubs of the human brain that diversively connect with multiple brain regions (van den Heuvel and Sporns, 2013; Tomasi and Volkow, 2011). Consequently, the formulation of this hypothesis should be motivated further.

We concur with the Reviewer that other regions, particularly those with diverse connection profiles, could a priori be expected to show more variable structure-function coupling. Although numerous evidence from macroscale cortico-cortical connectomics points to high participation coefficients in precuneus and posterior cingulate, we did not consider these regions as "exemplars" of this second hypothesis because:

- Previous studies did not take into account diversity of cytoarchitecture
- Previous studies did not take into account diversity of subcortical connectivity
- These regions are the focal points of the default mode network, which is among the most coherent networks and for which there is little evidence of dynamic reconfiguration

By contrast, insular cortex is cytoarchitecturally diverse, has extensive and diverse connectivity with multiple subcortical structure, and its affiliation with other regions/networks is relatively more fluid and less consistent (Uddin et al., 2019).

Following the Reviewer's suggestion, we have modified the Introduction to describe our second hypothesis more accurately:

"Another possibility is that structure-function coupling is greatest in regions that are intermediate in the putative unimodal-transmodal hierarchy."

Uddin, L. Q., Yeo, B. T. T., & Spreng, R. N. (2019). Towards a Universal Taxonomy of Macro-scale Functional Human Brain Networks. *Brain Topography*, 32(6), 926–942. <https://doi.org/10.1007/s10548-019-00744-6>

2- Regarding the possible effect of outliers on the metrics, the table with the robust estimation methods seems to indicate that degree, mean edge length and strength are the only metrics that do not change considerably. This should be discussed in the main text of the paper.

Following the Reviewer's suggestion, we have added the robust correlation analysis correlation coefficients and the three significant network features in the main text ("Results" section, "Spatial and topological determinants of dynamic structure-function coupling" subsection, paragraph #1):

"Robust correlation analysis (biweight midcorrelation and percentage bend correlation; Vallat 2018) suggests significant and stable correlations with structural degree (-0.1667;-0.1722;-0.1605), mean edge length (-0.1977;-0.1987;-0.1936), and functional strength (0.235;0.1956;0.1803)."

3- The authors used the difference between the 84th and 16th percentile, which corresponds to ± 1 standard deviation from the mean under normality. However, the paper does not demonstrate whether the data follows a normal distribution.

This is exactly why we used percentiles - so that we do not have to make assumptions about the functional form of the data distribution. For a normal distribution the standard deviation gives the interval where 68% of the data will fall. When the distribution is not normal, Q2 is the median, and we want 34% on each side, so we take the 16th and 84th percentiles.

4- The title is still very general and, in my opinion, justifying this choice in terms of a series of previous papers on the topic (which are also very general in my opinion) can be questionable. However, I understand your claim and I leave to the editors decide whether the title should be changed to describe the content of the work better.

We thank the Reviewer for their understanding, and we will respect the Editor's decision on this point. An alternative title could potentially be:

"Dynamic coupling of structural and functional connectivity in the brain"

or

"Time-resolved coupling of structural and functional brain networks"

REVIEWERS' COMMENTS:

Reviewer #2 (Remarks to the Author):

I thank the authors for addressing my comments. My revision is that the manuscript can be published in its current form.